# Experimental development of lightweight manipulators with improved design cycle time that leverages off-the-shelf robotic arm components

**Muhammad Rzi Abbas**[1], **Muhammad Ahsan**[1,2], **Jamshed Iqbal**[3]*

1 Department of Mechatronics and Control Engineering, University of Engineering and Technology, Lahore, Pakistan, 2 Human-Centered Robotics Lab, National Center of Robotics and Automation (NCRA), Pakistan, 3 School of Computer Science, Faculty of Science and Engineering, University of Hull, Hull, United Kingdom

* j.iqbal@hull.ac.uk

**Data Availability Statement:** All relevant data are within the paper.

## Abstract

The growing market for lightweight robots inspires new use-cases, such as collaborative manipulators for human-centered automation. However, widespread adoption faces obstacles due to high R&D costs and longer design cycles, although rapid advances in mechatronic engineering have effectively narrowed the design space to affordable robot components, turning the development of lightweight robots into a component selection and integration challenge. Recognizing this transformation, we demonstrate a practical framework for designing lightweight industrial manipulators using a case-study of indigenously developed 5 Degrees-of-Freedom (DOF) cobot prototype. Our framework incorporates off-the-shelf sensors, actuators, gears, and links for Design for Manufacturing and Assembly (DFMA), along with complete virtual prototyping. The design cycle time is reduced by approximately 40% at the cost of cobot real-time performance deviating within 2.5% of the target metric. Our physical prototype, having repeatability of 0.05mm calculated as per the procedure defined in ISO 9283:1998, validates the cost-effective nature of the framework for creating lightweight manipulators, benefiting robotic startups, R&D organizations, and educational institutes without access to expensive in-house fabrication setups.

## Section I: Introduction

The field of robotics has undergone a significant evolution, particularly with the emergence of lightweight robots, aligning seamlessly with the requisites of Industry 5.0 [1]. This progression has paved the way for substantial advancements in collaborative robotics, where lightweight robots, often referred to as "cobots," are tailored for harmonious human-robot interaction in applications ranging from industrial collaboration to medical-related scenarios [2–6]. These versatile entities boast characteristics that position them as adaptable collaborators within cooperative work environments. Their intrinsic attributes, characterized by reduced speeds and inertias, make them particularly conducive for seamless interaction with human

**Funding:** Grant Number: NRPU-16242 awarded by Higher Education Commission Pakistan. The funders had no role in study design, data collection and analysis, decision to publish, or preparation of the manuscript.

**Competing interests:** The authors have declared that no competing interests exist.

counterparts, leading to a broadening spectrum of applications across diverse domains. From assembly line installations [7] to medical setups [8], from education sector [9] to space endeavors [10], these robots have found their utility across varied industries. The escalating demand for automation positions lightweight robots as pivotal players across industrial sectors [1]. In this evolving landscape, our research takes on heightened significance, strategically addressing the imperative for streamlined design and development of lightweight robots, a cornerstone in facilitating the transition toward Industry 5.0-aligned industrial and R&D ecosystems.

Conventional engineering product design cycle [11] is illustrated in Fig 1 which is used to argue about robotic product development cycle. The whole process may be divided into seven major stages as labeled by circled digits in the figure.

- Stage 1 is the initial stage where the concept of the robot is formulated, and its basic design is outlined. It involves defining the robot's purpose, capabilities, and basic architecture. Typically, 10% [12–14] of the total design cycle time is spent in this stage.

- Stage 2 relates to the detailed design of the robot's components, such as mechanical, electrical, and software aspects. Planning for the manufacturing and assembly processes also occurs in this stage, and it takes around 20% of the total design cycle time.

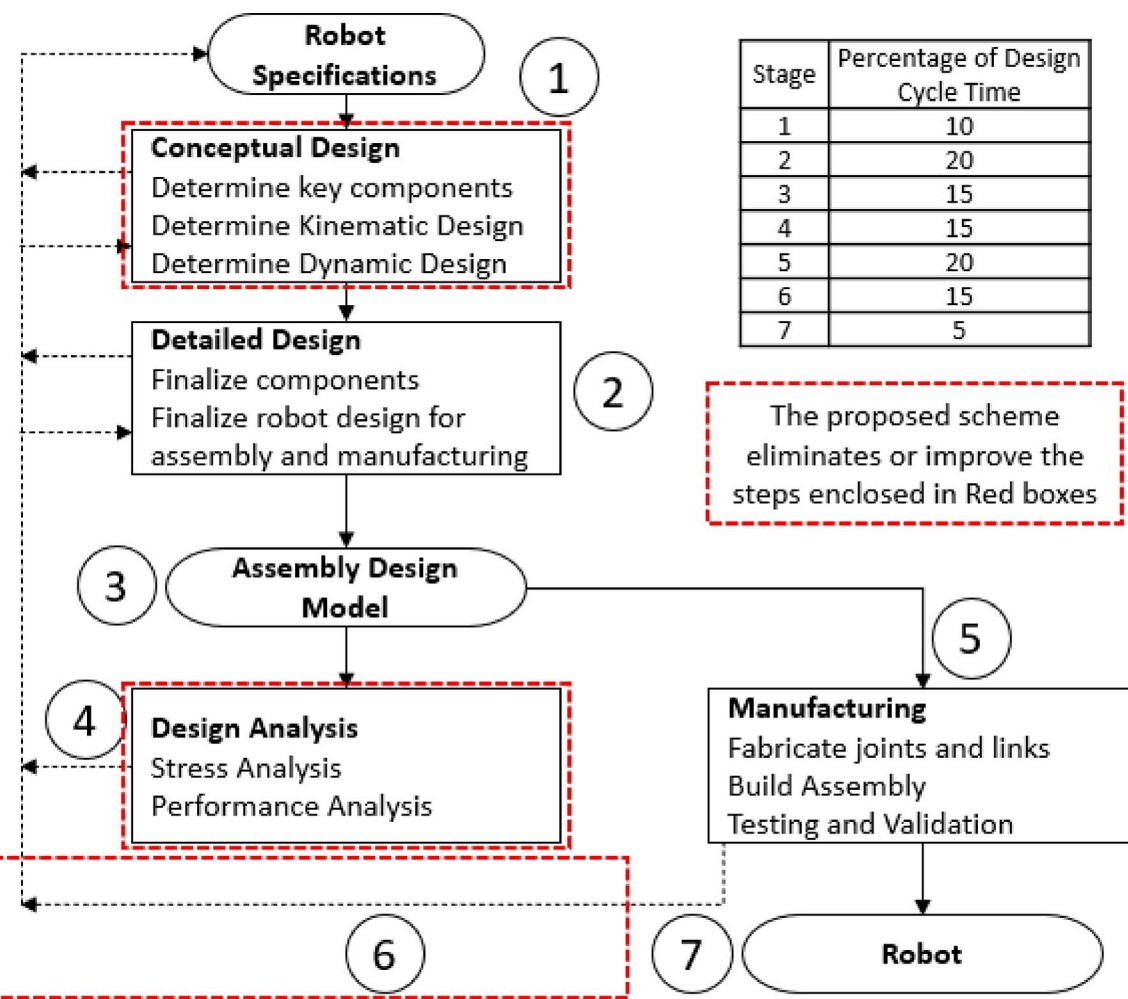

**Fig 1. Typical robot design and development cycle.** The table in the figure shows typically what percentage of total design cycle time [12–14] is spent on each stage. The portions enclosed in Red Boxes are either eliminated or simplified through the proposed scheme.

- Stage 3 implements the initial design, usually in simulated environments, which is a quicker method, or sometimes a physical prototype is also developed, that normally takes longer. Physical prototype development consumes around 15% of the design cycle time.

- Stage 4 tests and iterates the prototype based on several performance parameters. Typically, 15% of the design cycle time is spent on this activity.

- Stage 5 starts once the design is optimized and tested. The final version of the robot design is prepared for manufacturing. This includes sourcing components, manufacturing, and assembly. Normally 20% of the total design cycle time is utilized in this stage.

- Stage 6 tests and validates the manufactured robot to make it deployment ready. This stage typically consumes 15% of the total design cycle time.

- Stage 7 is related to production of documentation, user manuals, training material and deployment of the robot in the intended environment. This stage lasts around 5% of the total design cycle time.

The presented percentage values are based on general industry knowledge and practices. The breakdown of time spent on different stages of robot development can vary widely depending on the specific project, organization, and goals. These percentages are meant to give a rough idea of how time may classically be allocated in a typical robot development process.

Considering the extensive range of commercially available components, it would be both economical and timesaving for developers to simulate robotic manipulators exclusively composed of such components. The design approach presented in this paper modifies the robot development process along these lines, hence reducing the complexity and required time. First, stage 1 is reduced to only gathering of the robot specifications. Currently, these specifications are the degrees-of-freedom, payload capability, reach, and maximum workspace speed of the end-effector. The time needed to develop initial conceptual design based on the given specifications will be reduced because of the developed library of components. Once the library of subcomponents has been established, our proposed scheme will only use validated models of commercially available components to design the robotic manipulator. Hence stage 1 can almost be eliminated resulting in saving of around 10% of the total design cycle time. Second, the modeling of the subcomponents allows for virtual prototyping, eliminating the need for physical prototype development, saving an additional 10% of the total design cycle time by simplifying stage 2. Moreover, the developed library eliminates the requirement of subcomponents' performance analysis, further saving valuable development time. Lastly, the testing and validation of the manufactured robot is greatly reduced because the virtual prototype is made of all commercially available or produced components. Pearson's Correlation Coefficient and Root Mean Squared Error (RMSE) of trajectory tracking presented in the experimentation section of this paper, between the performance of the virtual prototype and the developed robot confirms this claim. Lastly, for the functional performance validation, our criterion requires that the physical prototype attains repeatability of the order of magnitude same as that of the commercially available lightweight robot. Table 7 in Section IV Hardware Experimentation validates that our developed prototype exhibits the desired performance.

This research work focuses on the simulation of hollow shaft Integrated Drive Joints (IDJs) integrated into lightweight robotic manipulators designed to operate as cobots. The developed library encompasses models of several commercially available subcomponents. Within the simulation environment, the IDJ models and the manipulator's structure are utilized to identify the suitable components, the frameless BLDC Motor and the strain wave gear, for the

integrated drive joints. Moreover, the simulation of the complete structure enables the consideration of coupled dynamics throughout the entire system, which is significant to control system engineers [15]. While a single IDJ exhibits distinct performance characteristics, coupling multiple joints introduces coupled dynamic effects across all joints [16, 17]. These couplings can be mitigated at the structural level through the appropriate selection of gearing systems or addressed at the software level by fine-tuning the control system accordingly.

The simulation tool was developed using the MATLAB/Simulink R2021b environment, while the robot structure was modeled using SolidWorks 2021. Subsequently, based on the findings obtained from the simulation tool, a 5 degrees-of-freedom (DOF) lightweight robot was developed, as depicted in Fig 2. Though the authors have also developed a 6-DOF lightweight robot by following the same footsteps, but for the sake of compactness of the article, the detailed analysis of 5-DOF robot is presented.

Table 1 summarizes the potential time savings offered by the proposed method for developing robotic manipulators. This data, based on research conducted at the Human Centered Robotics Lab, demonstrates a 38.6% reduction in the total design cycle time. The table itself is divided into six columns. The first two columns list the distinct stages involved in a typical engineering design process for a product. The third column displays estimated man-hours required to complete each stage under the conventional design approach, described in [11].

Eq 1, shown below, details the calculation used to determine these man-hour values.

$$Man\ hours = P\ x\ D\ x\ W \tag{1}$$

Where:

- P = Number of personnel assigned to a stage

- D = Number of days required to complete a stage

- W = Working hours per day

Column 4 details the actual man-hours consumed during the design and development of a 5-DOF lightweight robotic arm. Column 5 quantifies the man-hours saved by implementing the proposed approach, while Column 6 expresses these savings as a percentage. The sum of these percentages demonstrates the overall 38.6% reduction in design cycle time. It is important to note that the time distribution across different design and development stages may deviate from the general percentage values presented in Fig 1.

The organization of the remaining manuscript is as follows: Section II describes the challenges involved in the in-house development of light-weight robots and then propose a suitable methodology for generating an IDJ model that includes a synergetic combination of frameless brushless DC (BLDC) motor, strain wave gear and other subcomponents. Section III showcases the simulation results of the developed model, whereas Section IV details the performance results of the physically developed 5 DOF manipulator based on the in-horse manufactured integrated drive joint using the same hardware subcomponents as those used in the simulation. Lastly, Section V concludes the manuscript and steers the reader to possible avenues of future work.

## Section II: Addressing challenges involved in the in-house development of light-weight robots

Robot design methodologies encompass both bottom-up and top-down approaches, which have been extensively explored in literature. However, a critical gap remains: many analyses neglect the integration of commercially available, off-the-shelf components into the design

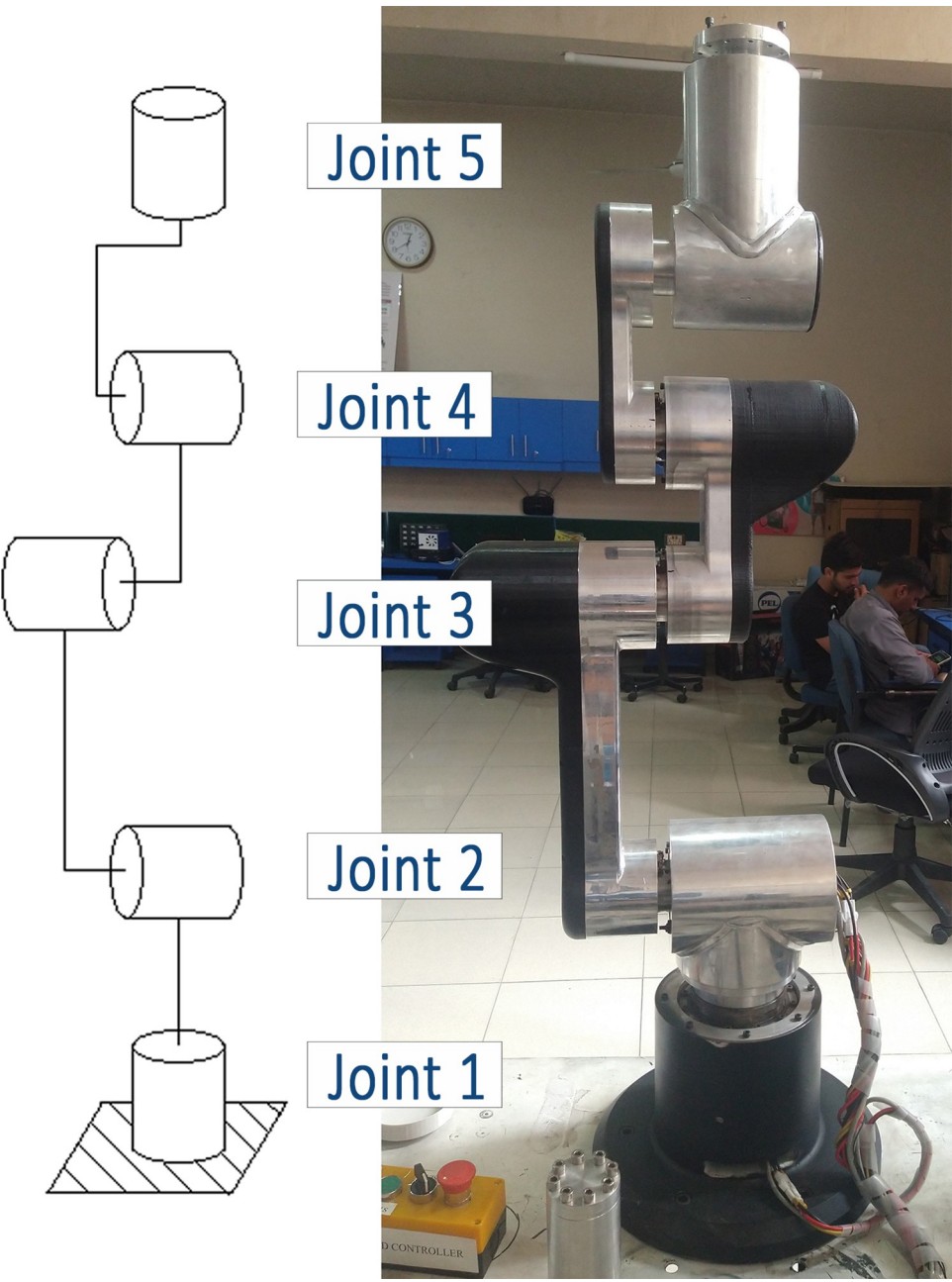

**Fig 2.** 5 DOF Robot's FBD (Left) and 5 DOF robot's hardware (Right).

process. Bottom-up strategies are well-suited for developing general-purpose robots with broad functionalities, while top-down approaches excel at creating robots specifically tailored for particular applications [18]. The top-down paradigm initiates the design process with a clear focus on the final application, resulting in a more streamlined design space. The recent studies [19] highlight the efficacy of top-down approaches in mitigating cyclic dependencies, particularly beneficial for the efficient design of mechatronic systems characterized by intricate interdependencies. Another relevant contribution to this field comes from a study that optimized reconfigurable collaborative robots, focusing on the importance of modularity and

**Table 1. Man-hours saved in result of following the proposed scheme for designing and developing a 5 degrees of freedom lightweight robotic manipulator in a typical R&D Lab.**

| Stage | Title | Conventional Design Approach's Initial R&D man-hour estimate | Actual R&D man-hours under the proposed scheme | Man-hours saved | Percentage of the Design Cycle Time saved |
|---|---|---|---|---|---|
| 1 | Conceptual Design | 160 | 32 | 128 | 5.9 |
| 2 | Detailed Design | 320 | 320 | 0 | 0.0 |
| 3 | Assembly Design Model | 480 | 480 | 0 | 0.0 |
| 4 | Design Analysis | 320 | 64 | 256 | 11.9 |
| 5 | Fabrication | 400 | 400 | 0 | 0.0 |
| 6 | Hardware Iterations | 480 | 0 | 480 | 22.2 |
| 7 | Commissioning | 80 | 80 | 0 | 0.0 |
| **Total Percentage of saved Design Cycle Time** | | | | | **38.6** |

structural design [20]. This innovative top-down approach draws inspiration from the V-model employed in the automotive and aerospace industries, as introduced in [21]. This optimization strategy aims to improve development cost-effectiveness by optimizing the physical structure, which in turn reduces manufacturing material costs. Additionally, leveraging established modular interfaces significantly cuts down on development time compared to customizing interfaces for each module, as detailed in [20]. These recent studies together highlight the changing nature of design processes. They reveal the valuable advantages of integrating off-the-shelf components within a top-down approach.

Like any robotic system, lightweight manipulators adhere to basic design principles and employ specific components to achieve their lightweight characteristics [22]. The IDJ or Direct Drive Joint (DDJ) incorporates frameless BLDC Motors and Strain Wave Gears, distinct from conventional manipulators. Additionally, lightweight manipulators utilize differently designed links made from lightweight materials. Over the past two decades, researchers have proposed various drive unit designs suitable for lightweight robots [22–25]. The design constraints imposed by the hollow shafts of drive units and of links connecting the drive units to form the kinematic chain, require specific component choices. For example, both the flexspline of the strain wave gear and the lightweight structure of the robot contribute to its overall mechanical flexibility, necessitating the consideration of dynamic coupling among multiple drive units through lightweight links. Consequently, optimizing the subcomponents of drive units and links becomes a primary objective in the designing and manufacturing of lightweight robots. Moreover, considering the extensive experience in the development of robotic manipulator components [26, 27], it is prudent to leverage this existing knowledge to narrow down the design space. Furthermore, researchers have also proposed schemes to optimize the selection of off-the-shelf motors, both AC Servos [28] and BLDC Motors [29], that can enhance the performance of already built manipulators. These studies analyze the dynamic behavior of the robotic system for optimizing the choice of motors using cost functions for energy consumption, trajectory tracking error, and total weight of the motors. Additionally, another research [30] showcases a method for testing drive-transmission couples suitable for following a defined motion. The analysis considers each pair separately without integrating them into a multi-DOF coupled system. Therefore, while this method enables the individual performance evaluation of drive-transmission couples, it does not provide analysis for situations where such pairs are joined together through a multi-DOF system. This highlights the need for comprehensive approaches that account for the dynamic interactions between coupled components within a multi-DOF robotic system.

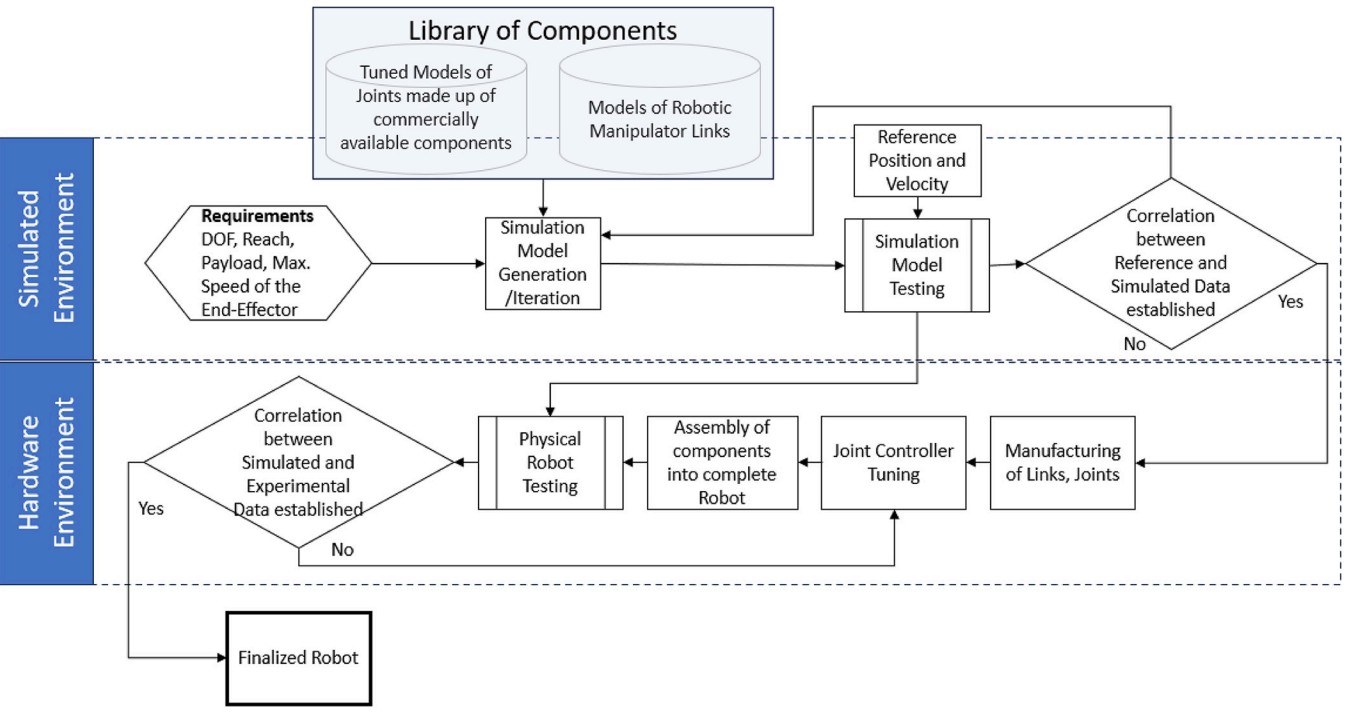

**Fig 3. Comprehensive workflow of the proposed scheme.**

Evaluating the performance of interconnected systems with multiple drives and links requires simulation tools. Unfortunately, current tools often lack the capability to simulate entire robotic systems with integrated drive joints, forcing developers to physically test and validate certain components. This gap arises because existing robot simulators typically allow designers to only simulate idealized systems or systems based on various assumptions [31, 32]. Modeling complex systems with readily available hardware is crucial. For example, when designing a speed reduction mechanism, engineers typically rely on calculations and simulations to identify suitable gears. However, these simulations might not account for real-world limitations like manufacturability or available parts. A more advantageous approach would be to explore commercially available gear options suitable for lightweight robots, model their performance, and then adapt other system components to achieve the desired speed reduction. This concept applies beyond gears–it extends to motors, sensors, bearings, motor drivers, and even the robot structure itself.

Industrial robotics has benefited for decades from the use of readily available components to build high-performance manipulators. A counter-intuitive approach, where existing components influence the design process, could be even more desirable. However, this would require easy access to comprehensive simulated models of all commercially available components. Unfortunately, the current landscape does not provide an extensive repertoire of simulated models, and even when such models do exist, there is a notable absence of a *unified and scalable platform* capable of testing the combinations of subcomponents employed in the fabrication of critical elements, such as a single joint within a robotic manipulator or the manipulator as a complete system made up of several subcomponents.

Fig 3 illustrates the comprehensive workflow of the proposed scheme. The end-user provides the initial specifications of the robot, which in the current stage are limited to the degrees-of-freedom, reach of the manipulator, payload carrying capacity, and the maximum

workspace speed of the end-effector. These specifications serve as the basis for component selection, leading to the creation of a Computer Aided Design (CAD) model of the robot. This CAD model is subsequently imported into the Simulink environment, where it is subjected to predefined trajectories. Based on the evaluation of position, velocity, and torque tracking performance, frameless BLDC motors and/or strain wave gears may be substituted, if necessary. Once satisfactory performance is achieved in the simulation, that is the robotic manipulator successfully follows the reference trajectories with correlation coefficient greater than 0.95, the robot is fabricated and assembled utilizing the commercially available components identified during the simulation phase. To complete the entire process, a comparative analysis is performed, contrasting the operational performance of the manufactured robot with that of the simulated robot. This research introduces a novel design and development framework for lightweight robots. However, opportunities remain for further optimization of each subcomponent, which will be a focus of future research.

## Methodology for implementation of the proposed scheme

Building a simulation model that captures all aspects of a real-world Integrated Direct Drive Joint (IDJ) is a complex task. However, the frameless BLDC motor and the gearing system emerge as the critical components influencing the IDJ's behavior. Therefore, detailed models were created for these elements, while less complex representations sufficed for smaller components like position encoders and bearings. This prioritization is justified because the BLDC motor and gearing system dominate the IDJ's inertia and electrical power consumption. Consequently, these two key components primarily define the IDJ's electrical and mechanical characteristics.

The development of models for the involved subcomponents necessitates consideration of several aspects. For instance, a motor requires modeling of both its electrical and mechanical attributes, whereas a gearing system pertains solely to mechanical aspects. However, different types of gearing systems exhibit distinct behaviors. For instance, planetary gear systems offer limited internal gear flexibility but have measurable backlash. In contrast, strain wave gears introduce flexibility through their flex spline and eliminate backlash. Backlash and rigidity are just two key differences between these gear types. Therefore, a comprehensive gear system model requires considering and optimizing various parameters based on specific needs. However, to simplify the process, focusing solely on modeling commercially available components' parameters is a practical approach, without concerning about the validity and optimization of such parameters. Consequently, developers can expedite the process by leveraging existing hardware components and utilizing their models within the simulation environment, effectively bridging the gap between simulation and actual hardware.

The subsequent discussion elaborates on the potential components that an IDJ may encompass. However, it is worth noting that certain components may be omitted in practice without compromising performance, albeit at the expense of non-performance-related features. Moreover, the simulated models utilize parameters sourced from component datasheets, which are typically freely available on the manufacturer's website.

### Frameless BLDC motor model

"Significant research already exists on simulation models for various motors, including BLDC motors. Since this study aims to find an effective combination of motor models, strain wave gears, and simplified models for other components, there's no need to develop entirely new models from scratch. Therefore, we leveraged standard BLDC motor models e.g. AC7 block of

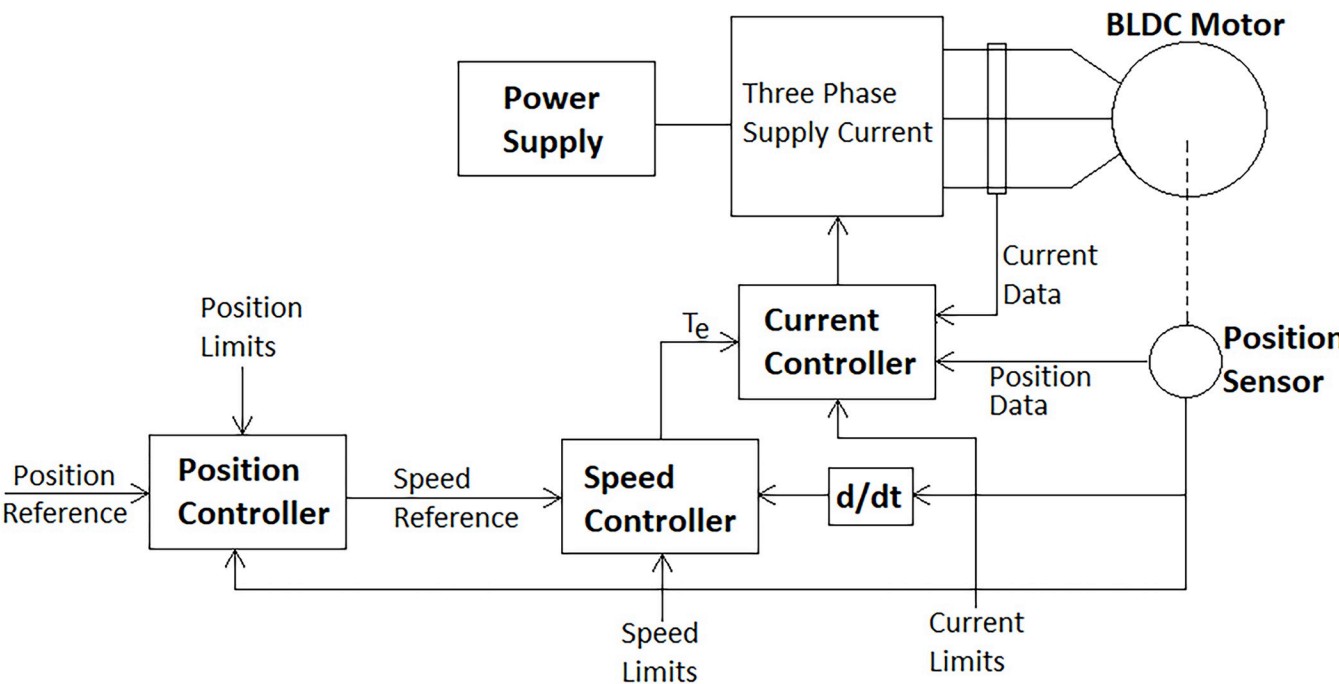

**Fig 4. General control scheme implementing closed loop control at all three stages, that is, torque (current), velocity, and position.**

Simulink MATLAB, and adapted them to create models for specific commercially available frameless BLDC motors.".

In addition to the BLDC motor, the drive system used to operate it plays a crucial role. The choice of drive system determines the required feedback and the level of control it can exert over the motor's motion. Typically, commercially available BLDC motor drives implement speed control through stator current control, employing a position sensor such as a rotary encoder or a hall sensor. The generalized mechanical system dynamics are represented by:

$$T_e = J\frac{d}{dt}\omega_r + F\omega_r + T_m \tag{2}$$

Where $T_e$ represents the electromagnetic torque, $J$ the inertia, $\omega_r$ the rotor speed, $F$ the viscous friction, and $T_m$ the load torque. These kinds of drives have an inner current loop that will take in the commanded torque and utilize the torque constant of the motor to generate suitable stator phase currents. Whereas the outer speed control loop takes in the load or applied torque and reference speed to figure out the electromagnetic torque as per the mechanical system's dynamic equation. Furthermore, commercial motor drives implement Proportional-Integral (PI), Proportional-Derivative (PD) or Proportional-Integral-Derivative (PID) controller for maintaining the desired speed-torque characteristics. These controllers are pre-tuned or may easily be tuned using the manufacturer's defined method and provided scheme. Another notable algorithm in this regard is the Sliding Mode Control (SMC). SMC is registered as one of the robust model-free control but it is complex in nature and the accompanied chattering limits its use in sensitive applications [33, 34]. Fig 4 shows the block diagram of the overall control scheme for commercially available motor drives.

This study employed BLDC motor controllers with PI controllers at all three levels: the current loop (innermost), velocity loop, and position loop (outermost). The integrator in PI controllers eliminates steady-state errors, which is particularly beneficial for lightweight robotic

manipulators. These manipulators often experience noisy signals due to weak sensor signals being close to powerful BLDC motors. Additionally, all commercially available hardware motor controllers used in this study relied solely on PI control, further supporting our choice. The PI parameters within these hardware controllers were adjusted according to the manufacturer's recommendations. and the trajectory following of a certain actuator before and after the tuning is shown in the hardware experimentation section, while the simulation-based motor controllers were autotuned using the damping factor and desired response time, as outlined in [35]. The damping factor is defined as follows:

$$\zeta = \frac{\omega_n K_p}{2K_i} \tag{3}$$

If $\zeta < 0.69$, then natural frequency is defined as:

$$\omega_n = \frac{-1}{\zeta \times T_{nl}} \log(0.05 \times \sqrt{1 - \zeta^2}) \tag{4}$$

Where if $\zeta \geq 0.69$, then natural frequency is defined as:

$$\omega_n = \frac{0.9257}{T_{nl}} e^{1.6341 \times \zeta} \tag{5}$$

## Strain wave gear model

In this research, the modeling of strain wave gears encompasses several crucial parameters that can be sourced from either the manufacturer's datasheet or the CAD model. These include weight, moment and product of inertia, stiffness, damping constant, nominal torque, and nominal speed. These parameters form the foundational framework for the accurate representation of strain wave gears within the simulation environment, ensuring a comprehensive and precise analysis of their performance characteristics.

While most parameters can be directly extracted, the moment and product of inertia are obtained from the manufacturer-provided CAD model. The damping constant is influenced not only by the grease used in the gearing system but also by the friction in the bearings and overall component alignment. Therefore, an empirical determination of the damping constant was performed for one of the manufactured joints to account for manufacturing imperfections and friction. Standardizing the manufacturing process, materials, assembly processes, strain wave gear grease, and bearings will improve the approximation of the damping constant. The determined damping constant was incorporated into the simulated model, present in the component library, to capture its effects.

## Robot structure

We modeled the robot structure with SolidWorks 2021. All necessary parts were meticulously modeled and analyzed for validity. The Integrated Direct Drive Joint (IDJ) was constructed using supplier-provided CAD models of its components: the frameless BLDC motor, strain wave gear, optical encoder, bearings, and the joint housing that encases them. Additionally, the links and connection flanges were modeled and subjected to stress-strain analysis to ensure the model's validity under worst-case loading conditions. Aluminum 6061-T6, known for its strength-to-weight ratio, was chosen as the material for the links and joint housings. Meanwhile, the connection flanges utilized high-strength Alloy Steel with a yield strength of $6.2 \times 10^8$ N/m$^2$. The factor of safety of the designed Aluminum links is at least 2.4, while for flanges its 4.3. Fig 5 shows stress analysis figure of one of the links being subjected to a maximum of $1.158 \times 10^8$ N/m$^2$ stress, while Aluminum 6061-T6 has yield strength of $2.75 \times 10^8$ N/m$^2$, hence

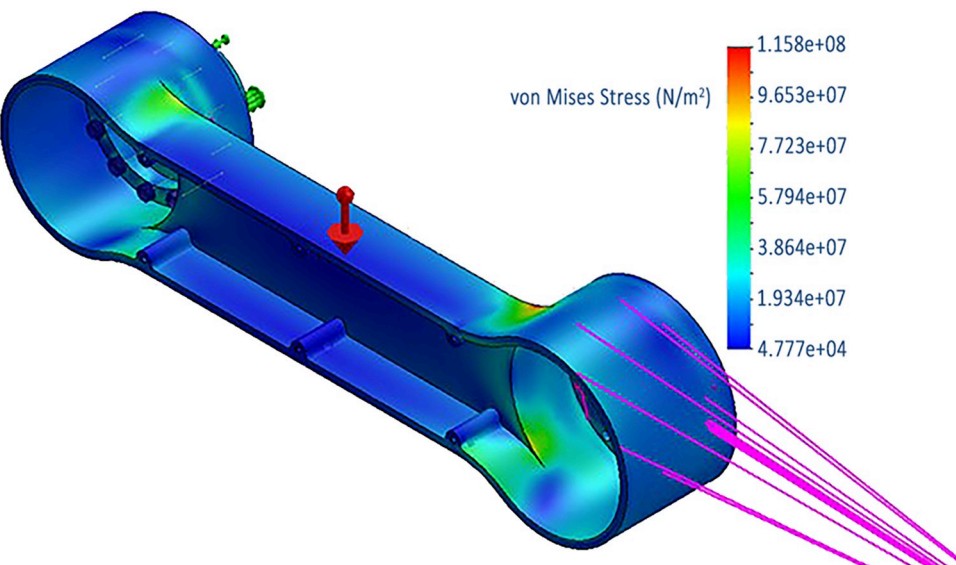

**Fig 5. Stress Analysis of one of the links showing maximum stress being applied on it in worst case.**

achieve 2.4 factor of safety. These values of the factor of safety ensure that the structure can easily withstand higher loading, hence increasing the structural integrity and life of the manipulator. Some covers which were not bearing any load, were later 3D printed, and were excluded from the analysis.

The CAD model of the robot's structure generated in SolidWorks was imported into MATLAB's environment for simulation. The CAD model considered the following parameters:

- Robot configuration

- Link weights

- Link moment and product of inertias

- Link lengths

Fig 6 illustrates one of the designed links and its integration with the joints to form the complete CAD model of the 5 DOF robot.

The integrated drive joint of the robotic manipulator includes additional subcomponents with minimal weight and moments of inertia. These components, listed below, can be grouped and represented as a single model by combining their net weight and inertia. Their individual influence on the overall dynamics is negligible due to their small size.

## Other components

Several other components, like miniaturized circuit boards, position sensors, bearings, torque sensors, brakes, and housings, might be included in an integrated drive joint [36]. These components typically have significantly lower weights and moments of inertia compared to the BLDC motor, strain wave gear, and robot structure. To simplify the analysis, these components are grouped together as a single unit. The next section details the considerations for modeling and manufacturing the integrated drive joint.

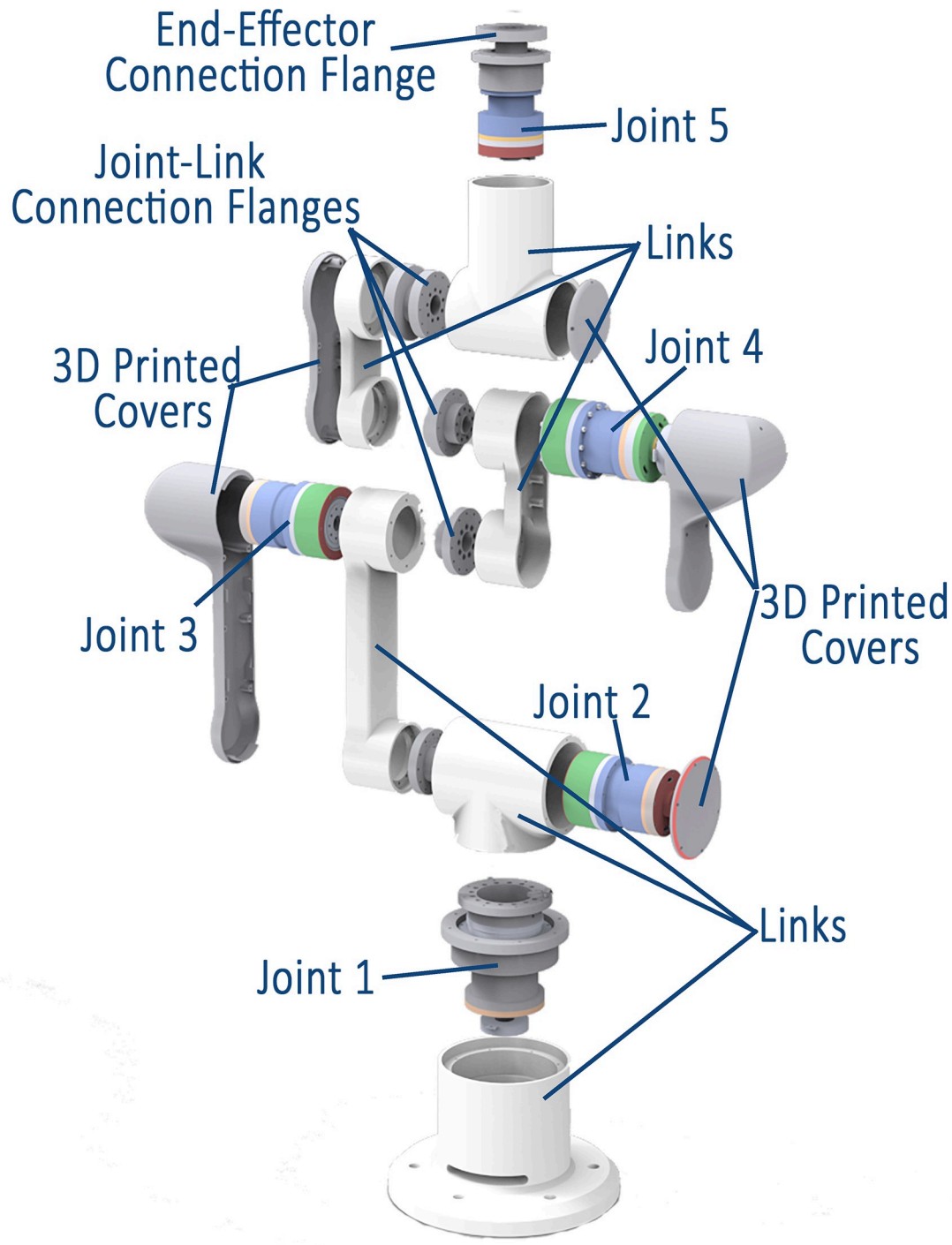

**Fig 6. Exploded view of the 5 DOF robot showing links, joints, connection flanges, and 3D printed covers.**

## Simulation model flow

The direct drive joint model was built by individually modeling and assembling commercially available subcomponents. These subcomponents include a frameless BLDC motor, strain wave gear, incremental position encoder, hollow shaft, bearings, joint housing, and various minor

**Table 2. List of BLDC motor parameters that were used for modeling the commercially available motors.** Sample of three different BLDC Motors are shown.

| Sr. # | Parameter | Unit | TMotor RI50 | TMotor RI80 | Maxon 588849 |
|-------|-----------|------|-------------|-------------|--------------|
| 1 | Resistance | Ohm | 0.673 | 0.196 | 0.109 |
| 2 | Inductance | mH | 0.474 | 0.364 | 0.0665 |
| 3 | BEMF Flat Area | rad | 1.197 | 1.197 | 0.761 |
| 4 | Torque Constant | mNm/A | 99 | 127.3 | 80.7 |
| 5 | Speed Constant | rad/s/V | 10.11 | 7.85 | 12.36 |
| 6 | Robot Inertia | g.cm$^2$ | 3500 | 6000 | 5300 |
| 7 | Poles | - | 7 | 7 | 11 |
| 8 | Nominal Torque | mNm | 500 | 1450 | 1010 |

parts. The electrical behavior of the BLDC motor was modeled in Simulink MATLAB using the Brushless DC Motor Drive (AC7) block, which simulates a standard current-controlled drive system for BLDC motors with closed-loop speed control. The mechanical aspects of the motor and strain wave gear were modeled within SolidWorks. The modeled parts make up the component library from which suitable parts are to be selected to make up the desired robotic manipulator. Table 2 presents the parameters used to model the frameless BLDC motor, extracted from commercially available datasheets. Furthermore, to account for potential manufacturing inaccuracies, up to 10% random variations were introduced in the weight, inertia, and stiffness parameters of the simulated direct drive joint and robot's structure. These variations compensate for possible deviations that may arise during the actual manufacturing process, such as material and machining differences and assembly imperfections. Additionally, an empirically determined damping constant was incorporated to model frictional forces. A test joint was manufactured to evaluate manufacturing imperfections and determine frictional losses, which were then included in the simulated models present in the developed library. All joints of the 5-DOF manipulator were configured with a damping constant of 0.035 Nm/rad/s. Fig 7 illustrates a snapshot of a test joint.

The developed simulator includes eight different frameless BLDC motors from T-Motor and Maxon, allowing for modeling of various motors without manufacturer dependence.

Similarly, several parameters of strain wave gear were used to model its effect in the direct drive joint, out of which the gear ratio is defined in the motor block. Apart from gear ratio, the weight, moment and product of inertias, stiffness, nominal torque, and nominal speed parameters were defined and used in different blocks. Furthermore, the simulated direct drive joint model includes the weights and inertias of position encoder, bearings, joint housing, and the hollow shaft through which the power and communication wires are routed.

Fig 8 shows the section view of the CAD model of the simulated direct drive joint.

Later, the simulated integrated drive joint was installed in the simulated robot structure, and the manipulator was driven on a defined trajectory. The position, velocity, and torque requirement graphs were plotted to establish the suitability of the simulated direct drive motor joint.

## Manipulator model

To evaluate the simulated integrated drive joint, a robotic manipulator with specifications defined in the Simulation Experimentation section was simulated. The development of the manipulator's model considered critical parameters, including robot configuration, link weights, link moment, product of inertias, and link lengths. These considerations were pivotal in ensuring the accuracy and effectiveness of the simulated manipulator's performance under various conditions.

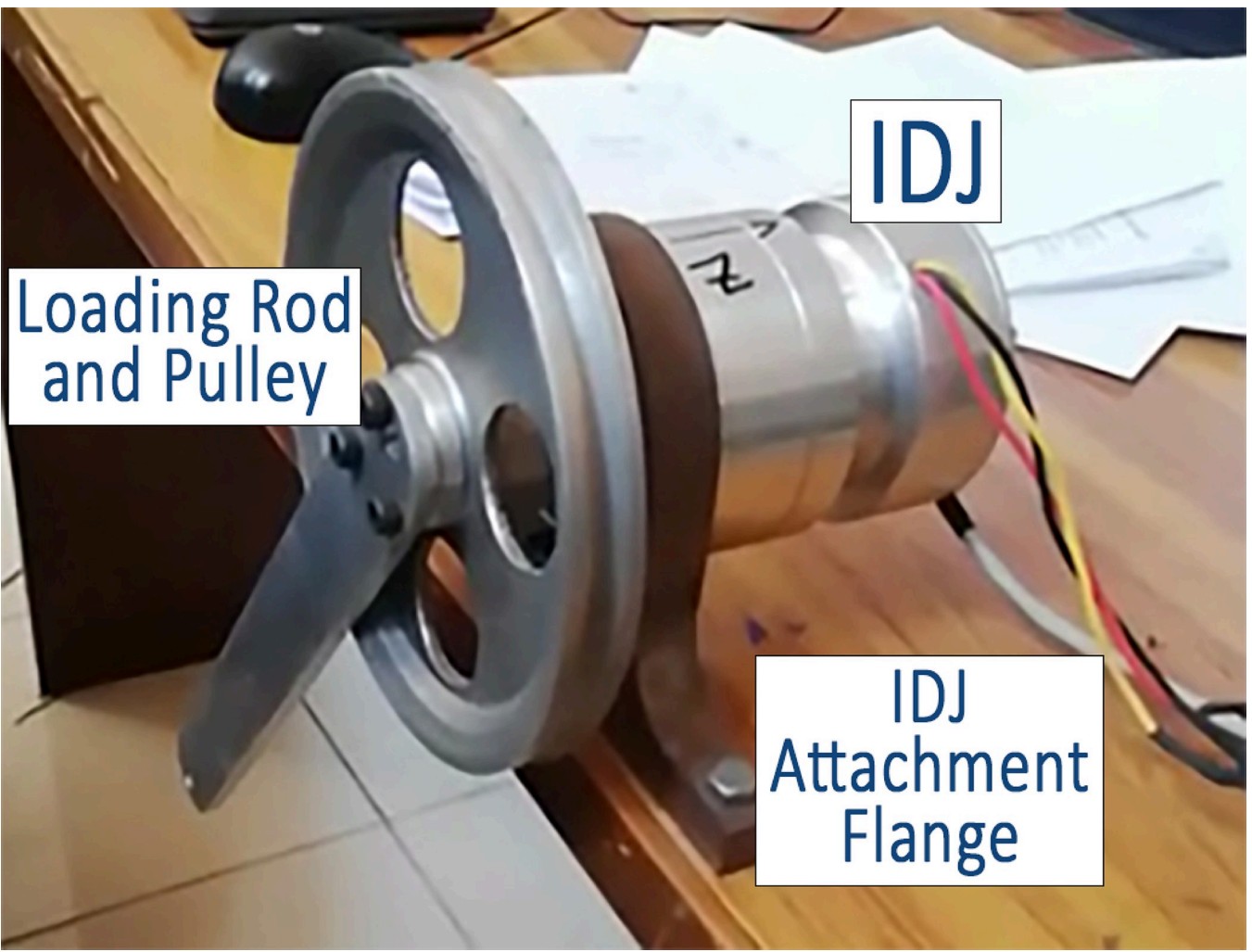

**Fig 7. IDJ testing setup.**

Table 3 presents the parameters of the robot links and joints used to generate the simulation model. These parameters were obtained from the CAD model, which considers the commercially available material, its properties, and the machining processes involved. While importing the model from SolidWorks 2021 to the Simulink environment, the dimensions and shape of the links were preserved. However, the shape and size of the joints were approximated by a point mass while maintaining all other parameters.

Individual links of the robotic arm were imported into the Multibody environment and were assembled in the defined configuration to build the complete structure. Fig 9 shows the Simulink model for building the robotic arm in MATLAB.

## Section III: Simulation experimentation

Using the steps outlined in Fig 3, the experimentation was initiated by specifying the initial specifications of the desired robot. The DOF were chosen to be 5, reach of about 1.1 m, with a payload capacity of 3 kg and as we are only focusing on developing lightweight robots then maximum speed of the end-effector was confined to 1 m/s. This value of speed ensures that the designed robot doesn't breach the safety guidelines laid down in ISO/TS-15066 [37]. In the

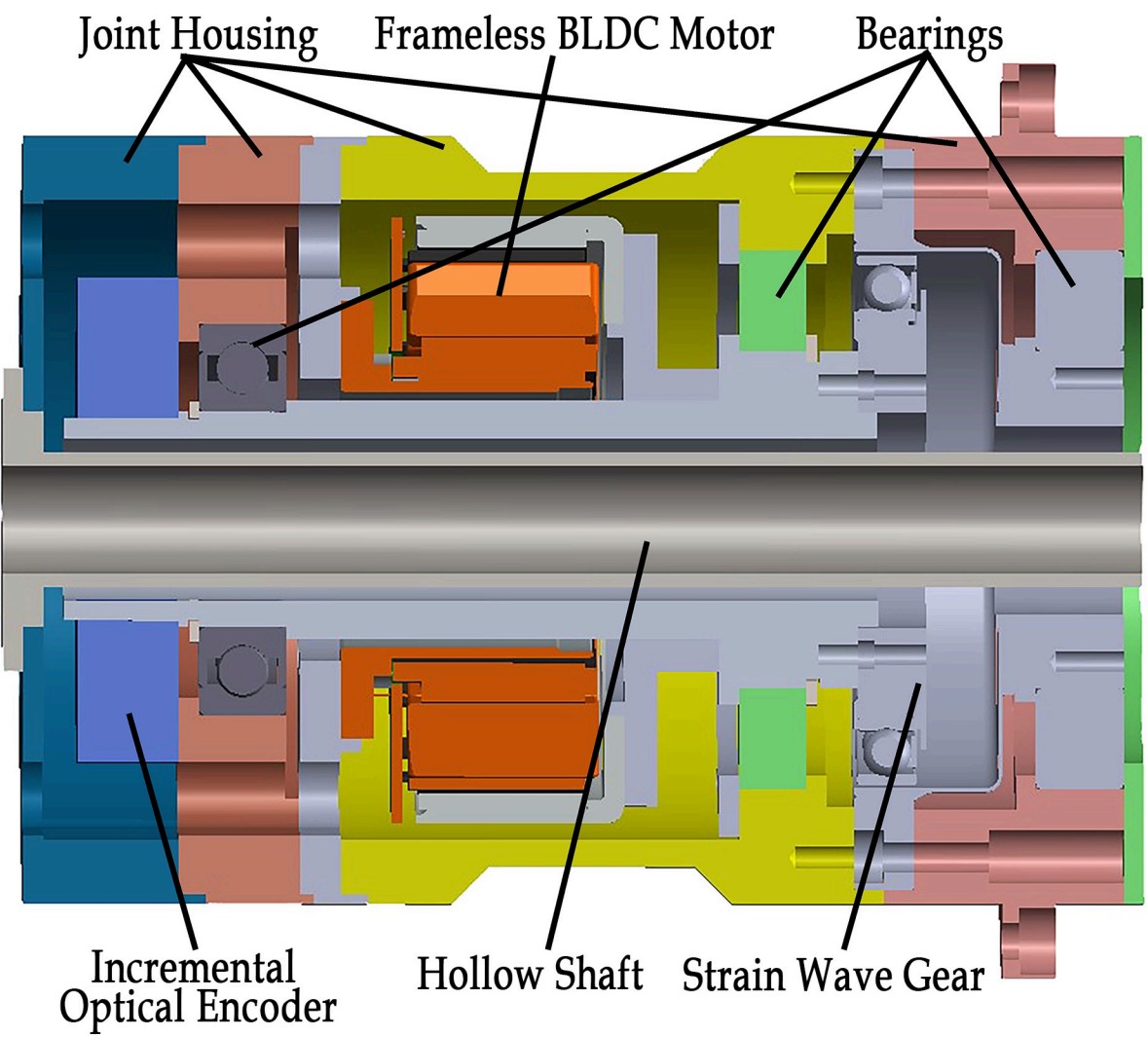

**Fig 8. Section view of the CAD model of the assembled Integrated drive joint.**

experimentation phase, direct drive joints featuring various commercially available frameless BLDC motors were integrated with the 5-DOF manipulator model made of manually selected robot links from the produced library. Trapezoidal velocity trajectories were generated as reference trajectories for different scenarios, and the simulated model was driven on these trajectories to evaluate the performance of the direct drive joints. Fig 10 provides a snapshot of the simulation running on a specific trajectory.

**Table 3. List of parameters used for modeling the structure of the robot in Simulink environment.**

| Sr. # | Manipulator Link | | Manipulator Joint | |
|---|---|---|---|---|
| | Parameter | Unit | Parameter | Unit |
| 1 | Size and Shape | mm | Point Mass | - |
| 1 | Mass | kg | Mass | kg |
| 2 | Center of Mass | mm | Center of Mass | mm |
| 3 | Moments of Inertia | g.cm$^2$ | Combined Moments of Inertia | g.cm$^2$ |
| 4 | Products of Inertia | g.cm$^2$ | Combined Products of Inertia | g.cm$^2$ |

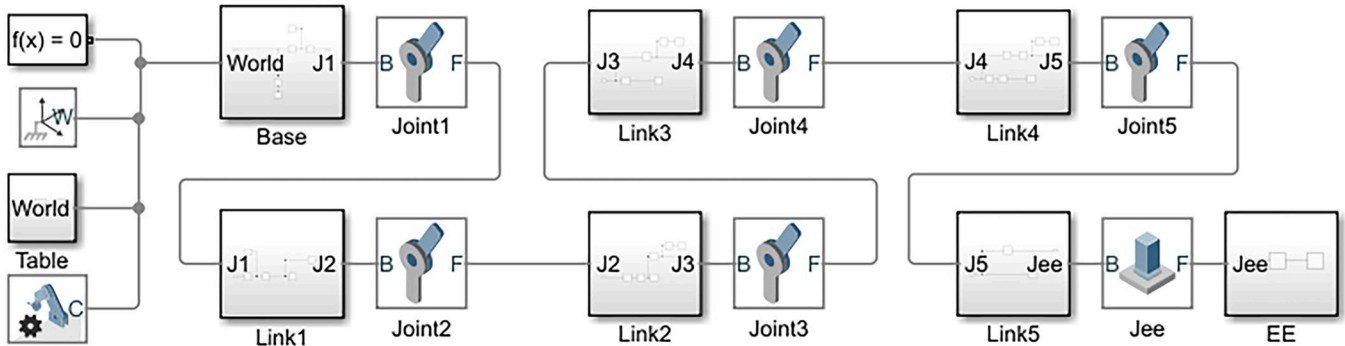

**Fig 9. Simulink SimScape Multibody model for building 5 DOF robotic arm.**

The simulation was conducted using the finalized parameters and SimScape Multibody model. The SimScape Multibody simulates system dynamics using a range of solvers, which can be variable or fixed steps. As we are going to test the physically manufactured robot on EtherCAT protocol, we used a fixed step solver with step-size of 0.1 ms. The EtherCAT protocol runs at a sampling time of 1 ms, that is why we had to run the simulation at an even smaller time step. The type of the solver was left to Simulink by allowing it to automatically select the

**Fig 10. Snapshot of simulation of 5 DOF manipulator on a given trajectory.**

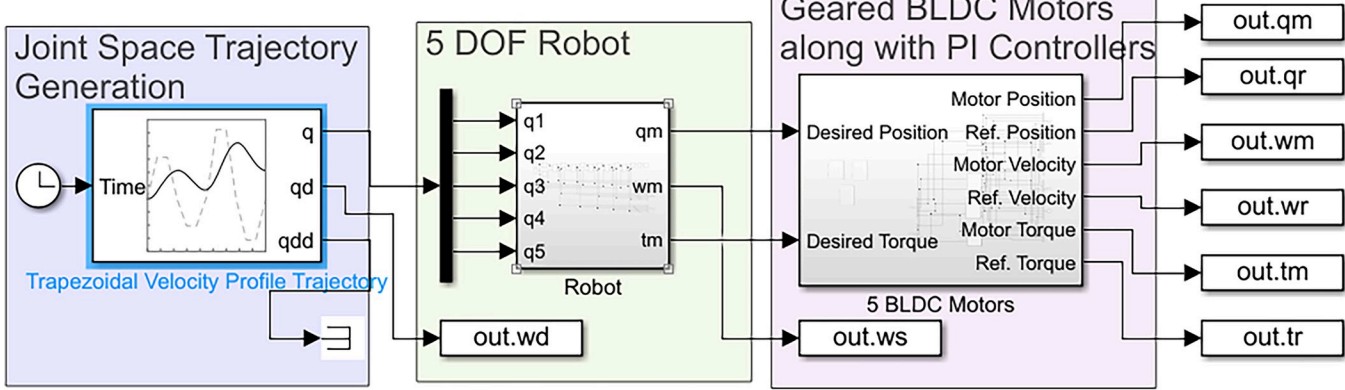

**Fig 11. System level Simulink model for simulating the robotic manipulator and its joints.**

solver type. The results aimed to verify if the simulated joints could precisely track the desired reference trajectories. For a robotic manipulator with interconnected joints and links, evaluating the coupled dynamics of the entire structure provided a more meaningful assessment than analyzing individual joint performance. In industrial settings, robotic manipulators encounter various disturbances from environmental interactions, unmodeled dynamics, and gravitational effects [38]. These disturbances are addressed at the system level. Fig 11 illustrates the system-level Simulink model used to simulate the complete robotic system.

The system model shown in Fig 11 outlines the simulation flow. A reference trajectory was generated in joint space and applied to the Multibody model of the robot. The model followed the trajectory, while position, velocity, and torque sensors captured corresponding signals. The sensed positions and torques were then used to control the BLDC motors through a PI-based control system. Joints that were capable of accurately following the desired position and torque profile exhibited minimal deviation between the output and reference signals. To quantitatively assess the similarity between the reference and simulated data, Pearson's Correlation Coefficient was calculated using the provided equation. Additionally, visual comparisons were also conducted.

$$r = \frac{\sum (x_i - \bar{x})(y_i - \bar{y})}{\sqrt{\sum (x_i - \bar{x})^2 \sum (y_i - \bar{y})^2}} \tag{6}$$

Where $r$ = Pearson's Correlation Coefficient, $x$ = value of the first variable, $\bar{x}$ = mean value of the first variable, $y$ = value of the second variable, and $\bar{y}$ = mean value of the second variable.

The Pearson's Correlation Coefficient remains a crucial metric to assess the degree of similarity between the reference and simulated torque profiles. It serves as an indicator of the suitability of the direct drive joint at its installed position. A low correlation coefficient value suggests an unsuitable joint, while a value closer to one indicates that the joint is well-suited for its installed position within the robot. The correlation coefficient is specifically calculated for the torque profile, as the correlation coefficients of the position and velocity profiles are dependent on the coefficient of the torque profile. Fig 12 demonstrates the response of an underpowered joint, highlighting the significance of adequate power for joint performance.

The underpowered joint may cause significant deviations in position and velocity responses from the reference trajectory due to insufficient torque. The generated torque fell short of desired value, reaching around 50 Nm compared to the required 70 Nm, leading to a failure in following the trajectory. The correlation coefficient of 0.2593 quantitatively indicated a

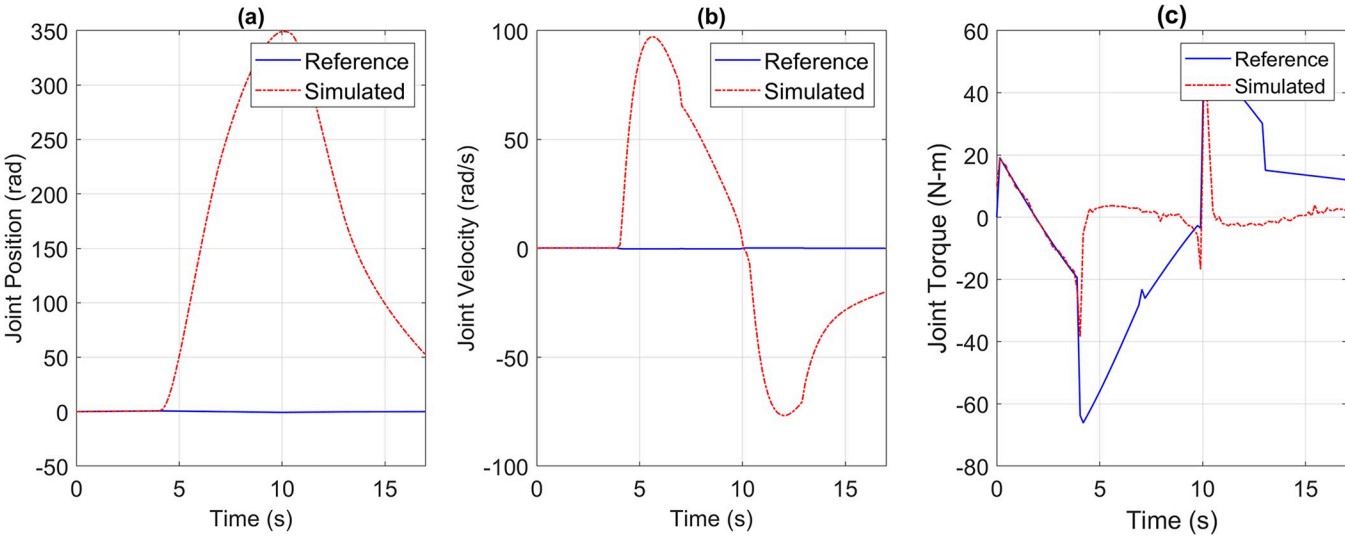

**Fig 12.** (a) Position, (b) Velocity, (c) torque reference vs simulation data of Joint 2 when an underpowered joint was installed at this location.

mismatch between the required and generated torque. To address this issue, we iterated the simulation using different integrated drive joints, gradually moving from low to high power joints. This IDJ selection method is currently manual with the aim of being automated in future. Currently the criteria for selection were to minimize the total weight of the joints and achieve enough torque capability to drive the robot on the reference trajectory with a correlation coefficient greater than 0.95. The selection process began with the last joint and moved backward, considering the impact of later joints on the earlier ones. This selection methodology helps identify optimal joints for each position. Intelligent selection methods will be implemented as the library of simulated integrated drive joints expands, currently including frameless BLDC motors and strain wave gears from Maxon, TMotor, and Laifual Drive. Furthermore, we successfully resolved the underpowered joint issue by substituting it with the suitable direct drive joints and tested the robot on a pick-and-place trajectory. Trajectories for position, velocity, and torque were obtained for all five joints of the 5-DOF robot. Through visual inspection, point-by-point comparison, and Pearson's correlation coefficient analysis, we confirmed that the joint trajectories of the simulated model closely followed the reference trajectories, surpassing the defined acceptable level of correlation coefficients greater than 0.95.

For the pick and place application, the reference versus simulated data for all the joints is shown in Figs 13 to 17.

The simulation results demonstrate that the simulated integrated drive joint successfully produces the reference trajectory in all cases. The torque graph reveals a disparity between the required and simulated torque, but this difference does not result in noticeable deviations in the position or velocity trajectories due to the robustness of the PI controller. Additionally, Table 4 provides the Pearson's Correlation Coefficients for all joints, quantifying the degree of match between the reference and simulated trajectories.

The high correlation coefficients obtained in the simulation experiments highlight the effectiveness of the simulation tool and the chosen motor and harmonic drive combinations in achieving accurate and reliable robot motion control. These results demonstrate that the simulated robot performance closely aligns with the intended reference trajectories, which is crucial for ensuring precise and consistent robot motion.

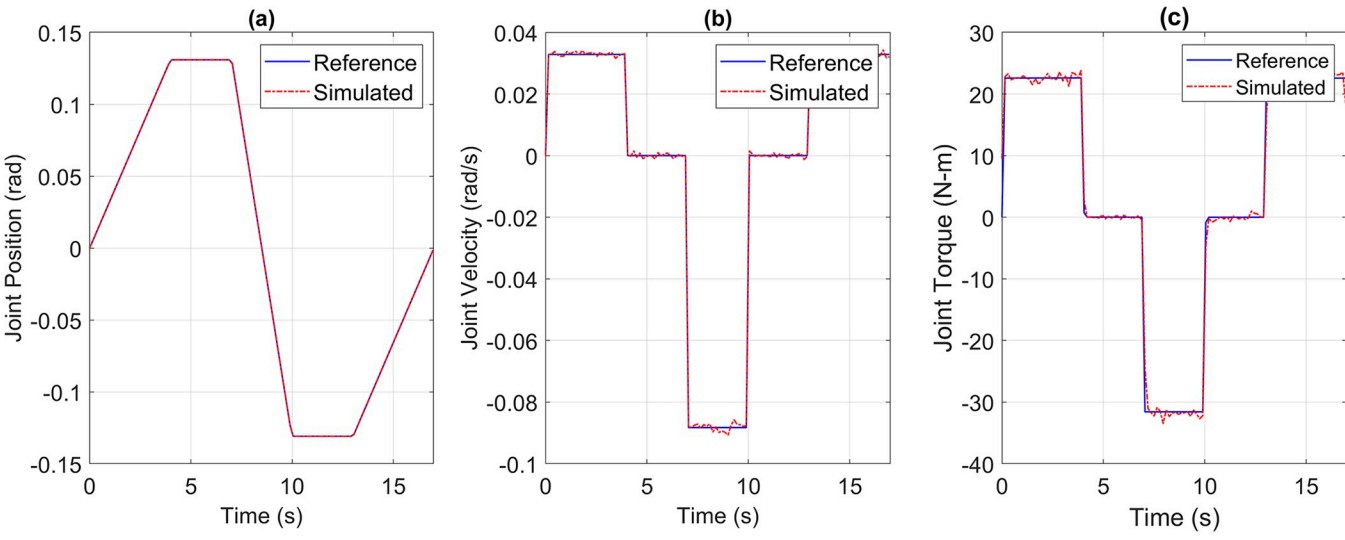

**Fig 13.** Reference vs Simulation Data of Joint 1's (a) Position, (b) Velocity, and (c) Torque.

Several sources of noise and disturbances including EMI, sensor (encoder) noise, input and output disturbances can cause the manipulator to deviate from an ideal trajectory in real-life. We analyzed how adequately our PI controller would reject sources of noise disturbances in real-life by lumping these at the output of the speed controller, in the form of a single noise source. Specifically, this is an additive white gaussian noise with unit variance, resulting in a signal-to-noise ratio (SNR) of 15 dB, which is in close agreement with (Fig 22) noise levels observed in real experiments. Fig 18 depicts the responses of position, velocity, and torque tracking for one of the joints, joint 3, subject to noise/disturbance.

In Fig 18, although there is distortion in the speed and, consequently, in the torque, the PI controller implemented in the position loop adeptly mitigates the effects of the introduced disturbance. This reinforces the robustness of the controller.

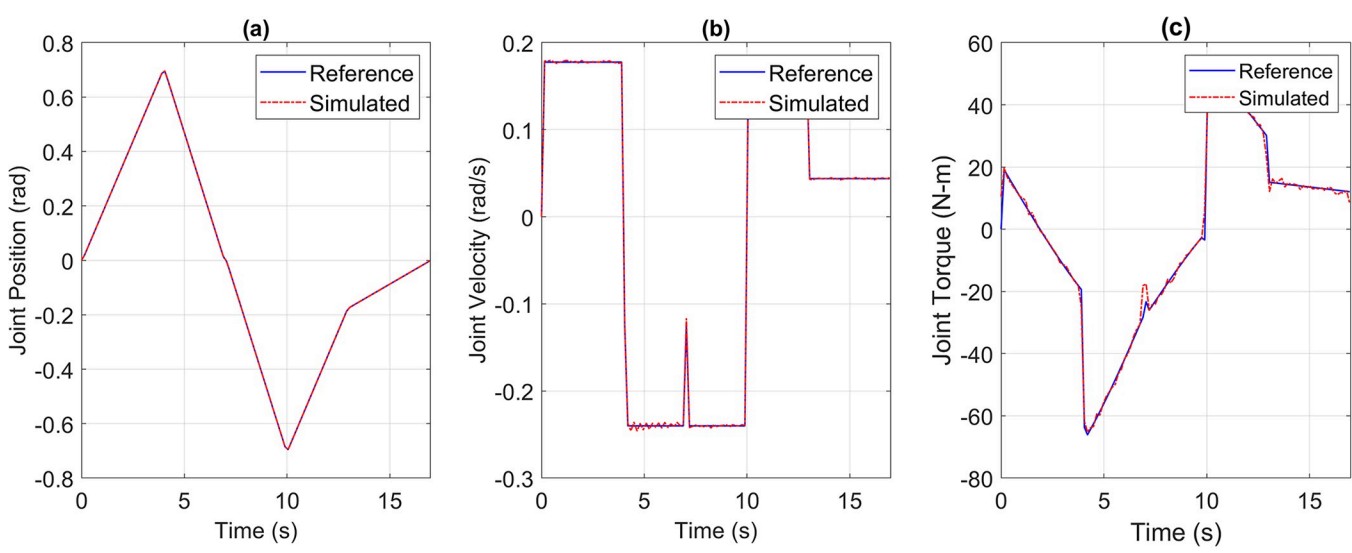

**Fig 14.** Reference vs Simulation Data of Joint 2's (a) Position, (b) Velocity, and (c) Torque.

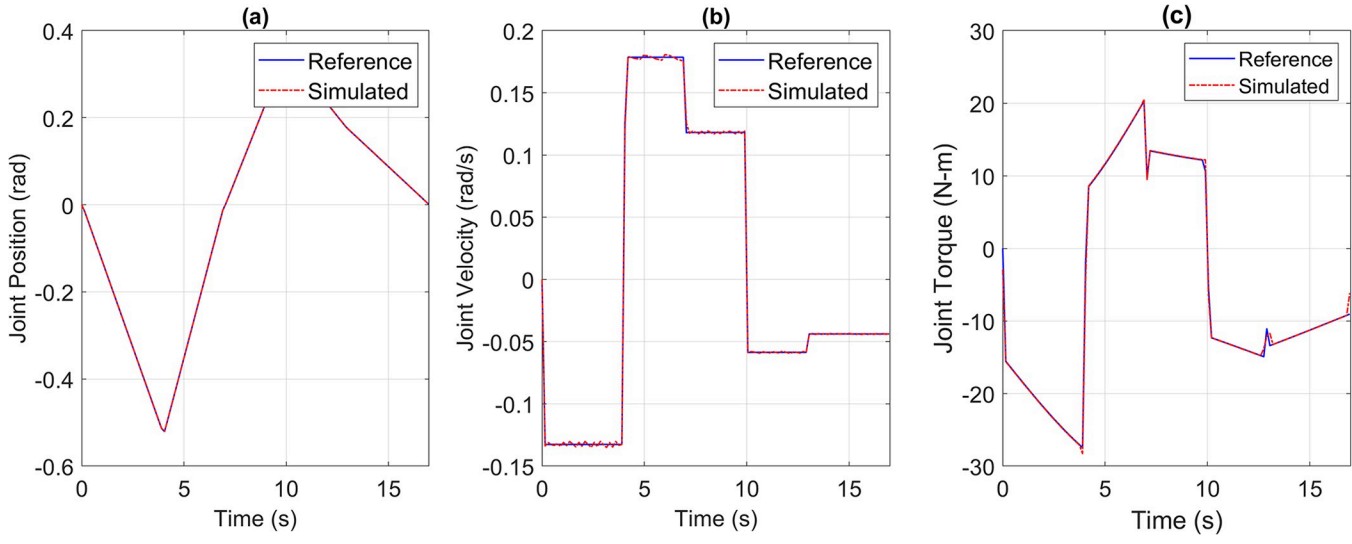

**Fig 15.** Reference vs Simulation Data of Joint 3's (a) Position, (b) Velocity, and (c) Torque.

This finding carries significant implications for practical applications, as it suggests that physically manufactured robots using the same integrated drive joints and control system can potentially achieve similar performance. The subsequent section presents the results of fabricating the robot based on the integrated drive joints recommended by the simulation studies.

## Section IV: Hardware experimentation

Based on the simulation results, integrated drive joints were built using the recommended commercially available components. To comply with modern industrial standards, Nanotec® BLDC motor controllers with EtherCAT communication capabilities were chosen to drive the BLDC motors. Five such joints were manufactured alongside the manipulator's structure. The

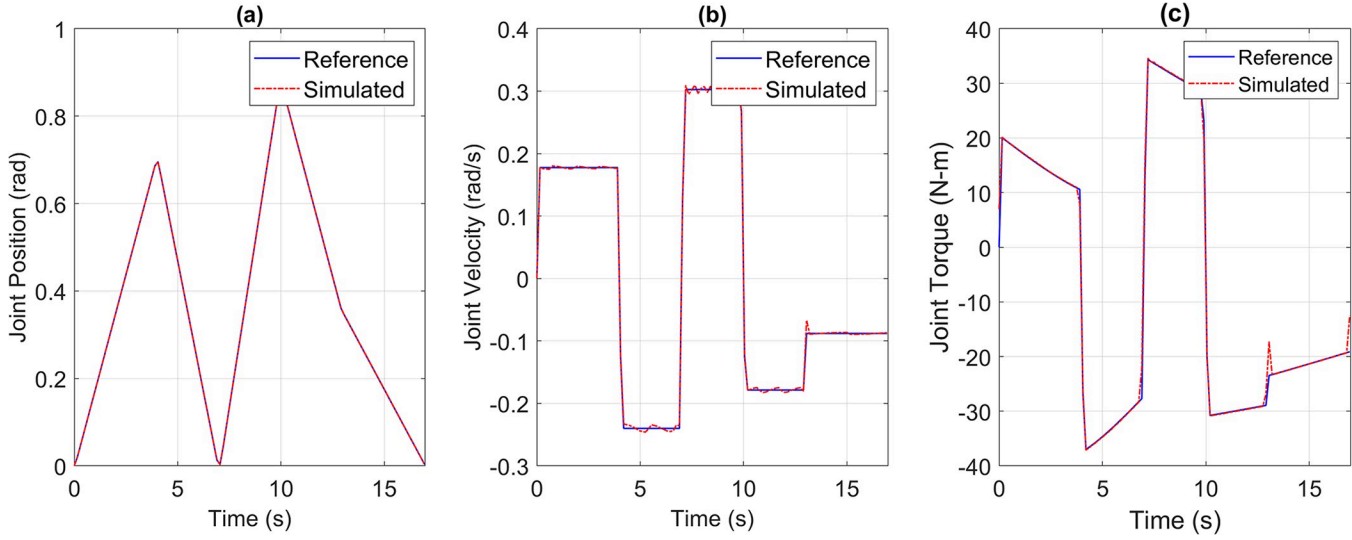

**Fig 16.** Reference vs Simulation Data of Joint 4's (a) Position, (b) Velocity, and (c) Torque.

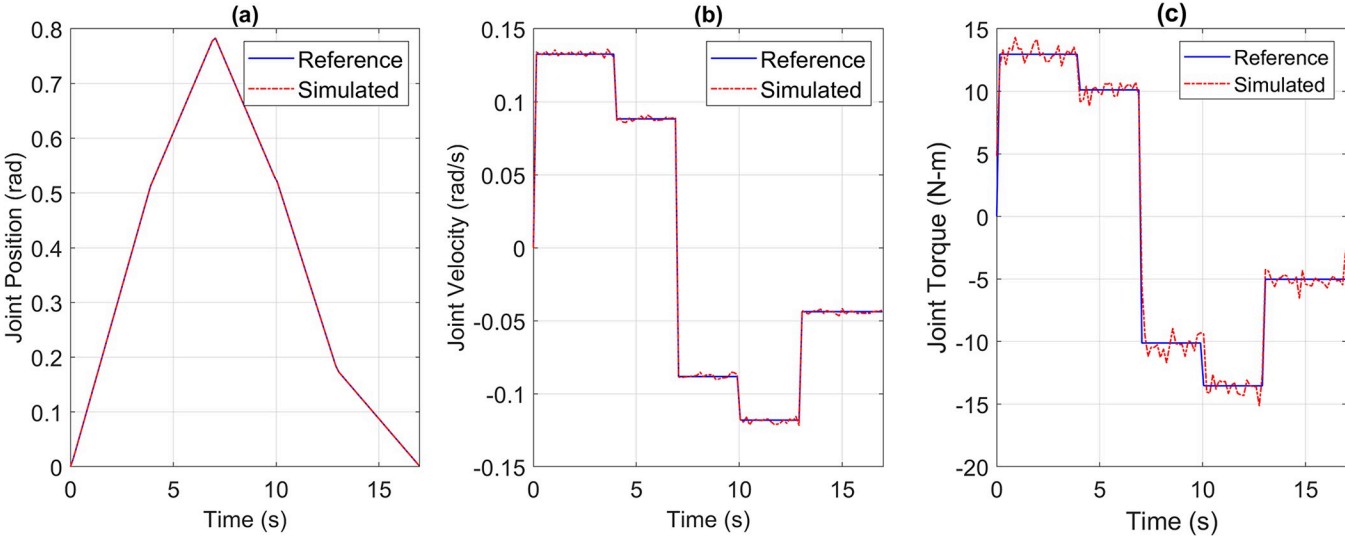

**Fig 17.** Reference vs Simulation Data of Joint 5's (a) Position, (b) Velocity, and (c) Torque.

complete assembly resulted in a 5-DOF manipulator, pictured in Fig 19, controlled via Ether-CAT-enabled BLDC motor controllers.

We used Plug & Drive software, provided by Nanotec®, for tuning the PI control loops of the motor controllers. This motor controller, similar to the ones used in the simulation, implements the PI control loops on three levels, that is, the inner most on torque, then on velocity, and the outer most on position. Moreover, a non-jerk-limited motion mode was used to implement the trapezoidal velocity profile motion. The acceleration and deceleration of all joints was limited to 200 rad/s². This value of acceleration limit was figured out keep in view the maximum current limits of the attached BLDC Motors. The position and velocity following response of one of the motor controllers before and after tuning is shown in Figs 20 and 21.

The physical prototype robot was programmed to follow the trajectory used in the simulation, and the corresponding torque, velocity, and position data from the integrated drive joints were collected. Joint angle limits were also coded into the controller to ensure movement of the robot is in the reachable free workspace. Table 5 gives the joint limits of all joints.

The collected data was compared with the simulated data for the pick and place joint space trajectory, as shown in Figs 22 to 26. It is important to note that the time constant of a geared joint prevents any noticeable movement of the output shaft due to current spikes. However, for experimentation purposes, the torque data was obtained from the current sensor in the motor controller, at a sampling rate of 1000 Hz, and passed through a moving average filter with a window size of 15 samples, approximately equal to 0.015 seconds in time. This moving average filter emulates the damping effect of the geared joint.

Once again, visual and Pearson's correlation coefficient were used to establish the similarity of the trajectories coming from the simulated model and the actual robot. Table 6 lists the Pearson's Correlation Coefficient for torque profiles of all joints.

**Table 4. Pearson's correlation coefficient between reference and simulated torque profiles of all joints for pick and place trajectory.**

| Parameter | Joint 1 | Joint 2 | Joint 3 | Joint 4 | Joint 5 |
|---|---|---|---|---|---|
| Correlation Coefficient | 0.9969 | 0.9984 | 0.9982 | 0.9992 | 0.9963 |

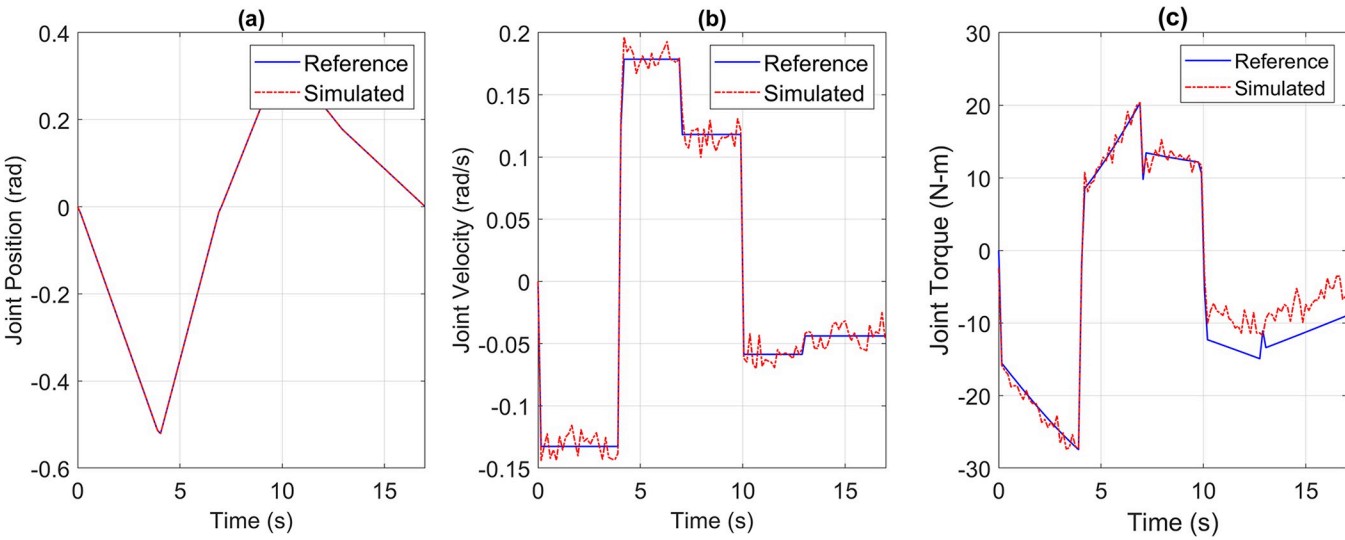

**Fig 18.** Reference vs Simulation data of Joint 3 after introducing disturbance, (a) Position, (b) Velocity, and (c) Torque.

Although the correlation coefficients have slightly decreased, they remain within an acceptable range as the position and velocity trajectories are still being accurately followed. However, the mismatches in the torque trajectories require further investigation. The torque impulses observed in the physical prototype come from the current sensing on the motor controller.

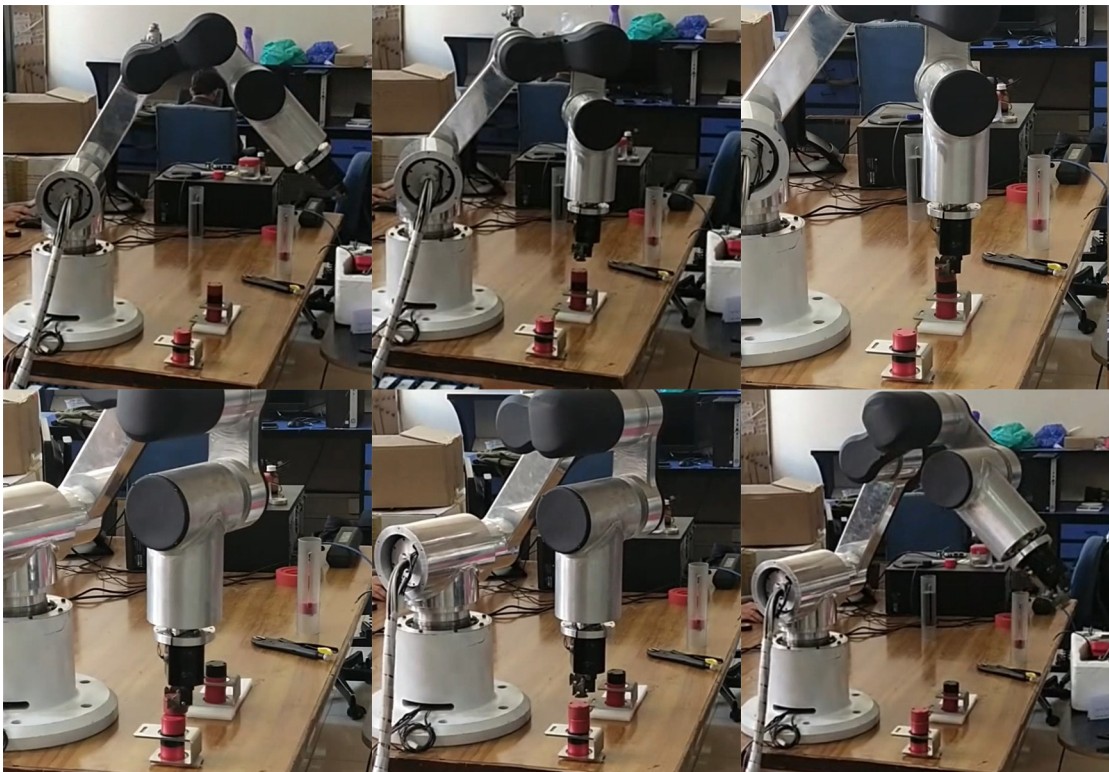

**Fig 19. Manufactured 5DOF robot with integrated drive joints performing a pick and place task.**

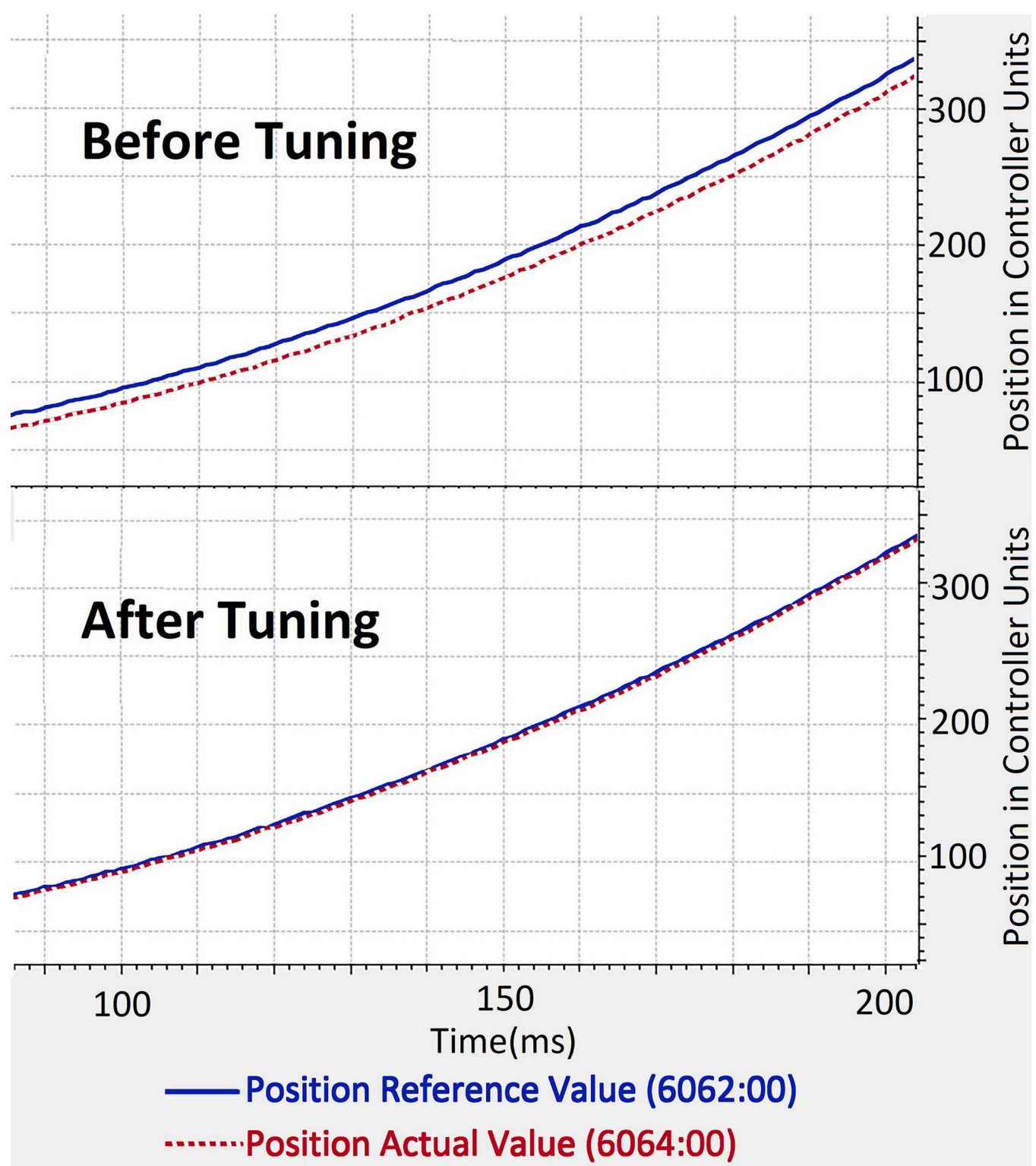

**Fig 20. Before and After tuning response of position (left) and Velocity (right) of one of the joints.** The x-axis shows time in milliseconds, whereas y-axis shows the position and velocity in motor controller units.

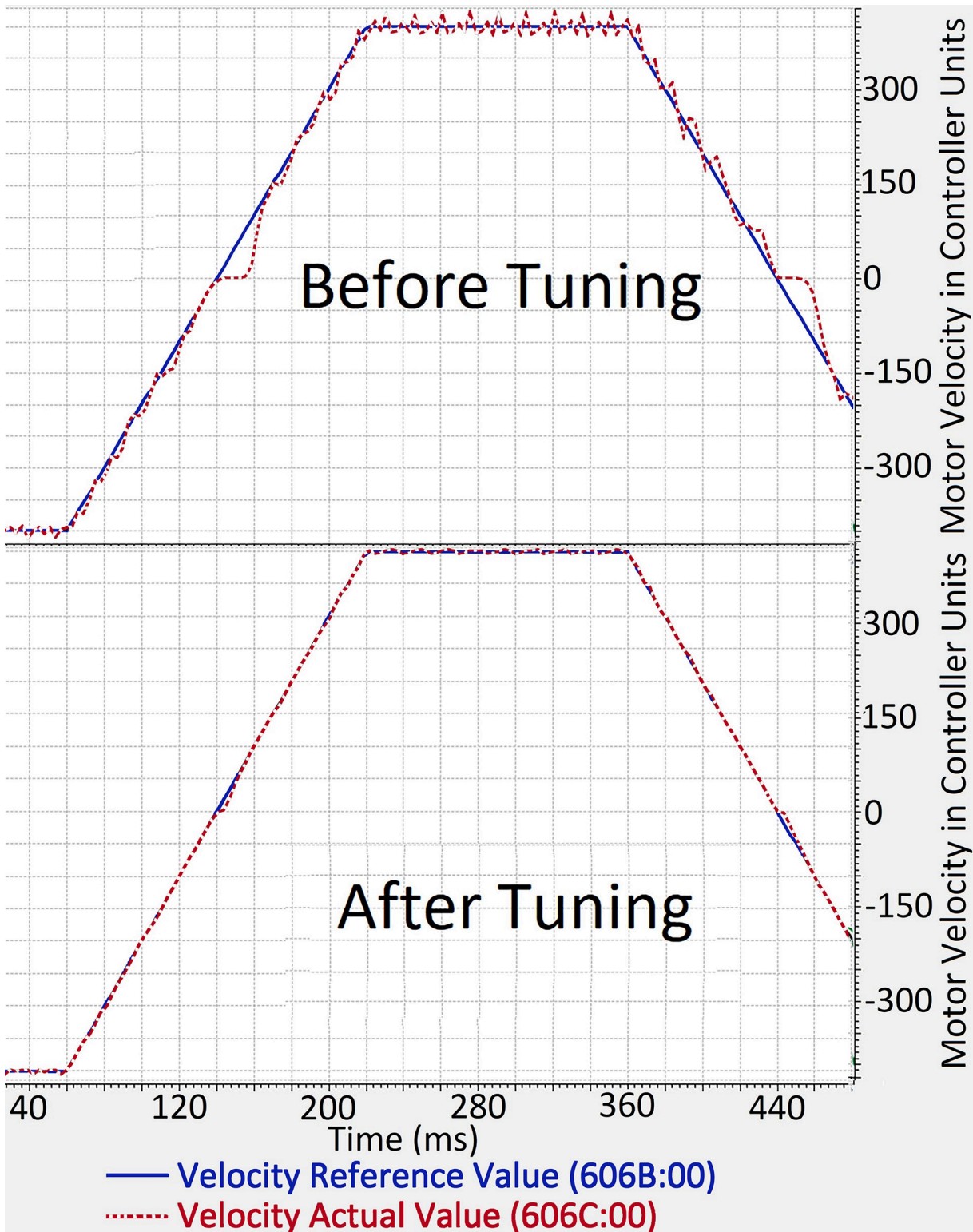

**Fig 21.** Before and After tuning response of position (left) and Velocity (right) of one of the joints. The x-axis shows time in milliseconds, whereas y-axis shows the position and velocity in motor controller units.

**Table 5. Joints angle limits (radians) for avoiding singularities.**

| Parameter | Joint 1 | Joint 2 | Joint 3 | Joint 4 | Joint 5 |
|---|---|---|---|---|---|
| Allowed rotation (rad.) | +/- 2.97 | +/- 1.57 | +/- 2.62 | +/- 2.62 | No limit |

These impulses are within the current limits of the BLDC motors and do not cause any deviation in the required position trajectory. Additionally, the average torque deviation from the simulated model remains below 5 Nm for most of the time, which is a small percentage of the nominal torque capability of the integrated drive joints. This additional torque compensates for unmodeled dynamics, friction, and manufacturing variations. Simulation tools capable of predicting the robot behavior in an ideal world may fail to account for the performance under real-world imperfections such as friction and manufacturing variations. However, the position and velocity following performance of the robot is still below acceptable level due to the disturbance rejection and noise cancellation properties of the PI controllers used. These results emphasize the importance of considering real-world complexities and the need for ongoing calibration and optimization. We aim to further investigate this problem to minimize the simulation-experiment performance disparity.

Apart from visual comparison and correlation coefficient between the simulated and the actual trajectories, trajectory tracking error was also calculated to further establish the trajectory following capability of the physically developed manipulator. We calculated Root Mean Squared Error (RMSE) for all five joints using Eq 6, given below.

$$RMSE = \sqrt{\frac{1}{N}\sum_{i=1}^{N}\left(x_{ref,i} - x_{measured,i}\right)^2} \tag{7}$$

Table 7 gives the RMSE of joint space position, velocity, and torque trajectories. Position and velocity RMSE signifies the position and time tracking efficiency of the developed robot, whereas torque RMSE once again suggests that the deviation in torque is a result of unmodeled dynamics. The maximum RMSE of torque is for joint 2, which is 5.4244 Nm, and if considered along with Fig 23, then this is around 10% of the total torque output for the desired trajectory. Moreover, the IDJ installed at joint 2 can produce torque up to 145 Nm, therefore, if the

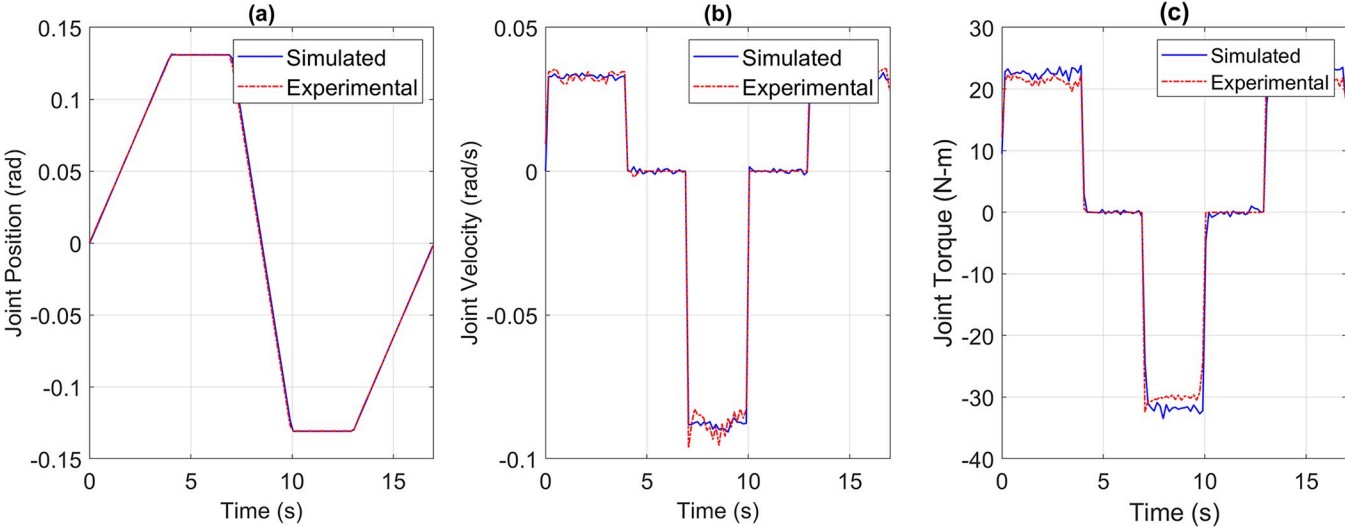

**Fig 22.** Simulated versus Experimental data of Joint 1's (a) Position, (b) Velocity, and (c) Torque.

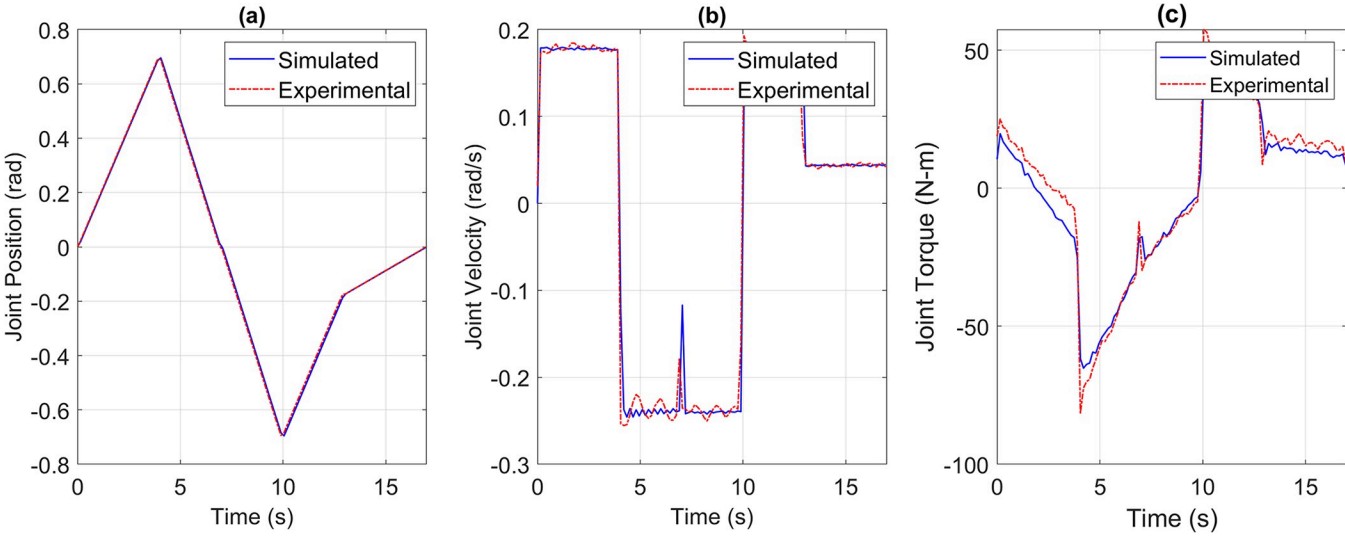

**Fig 23.** Simulated versus Experimental data of Joint 2's (a) Position, (b) Velocity, and (c) Torque.

RMSE is seen in comparison to the maximum torque capability of the joint then it is only around 3.74%.

Furthermore, the repeatability of the developed 5-DOF robot prototype was calculated as per the procedure described in ISO 9283:1998 [39], defined specifically for industrial robots. A dial gauge having positional resolution of 0.01mm was used to measure the position of the end-effector of the robot. Five different positions were commanded to the robot and 100 iterations were carried out for each position to generate the position data. To calculate the positioning repeatability the following calculations were carried out.

$$l_i = \sqrt{(x_i - \bar{x})^2} \tag{8}$$

$$\bar{l} = \frac{1}{n}\sum_{i=1}^{n} l_i \tag{9}$$

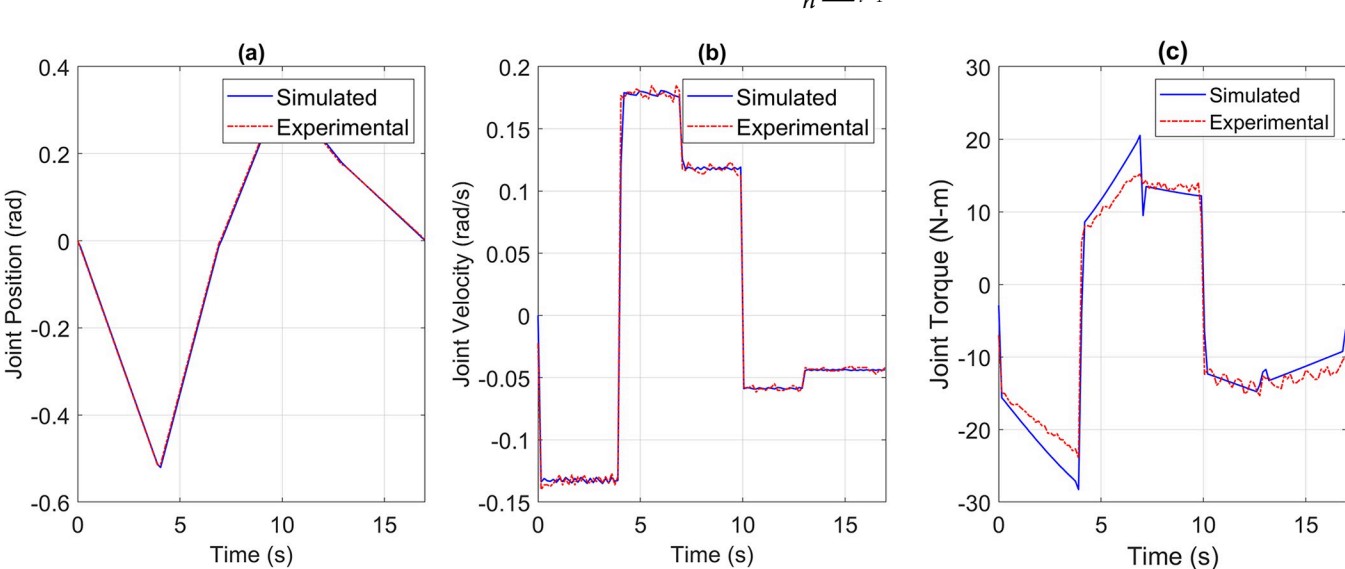

**Fig 24.** Simulated versus Experimental data of Joint 3's (a) Position, (b) Velocity, and (c) Torque.

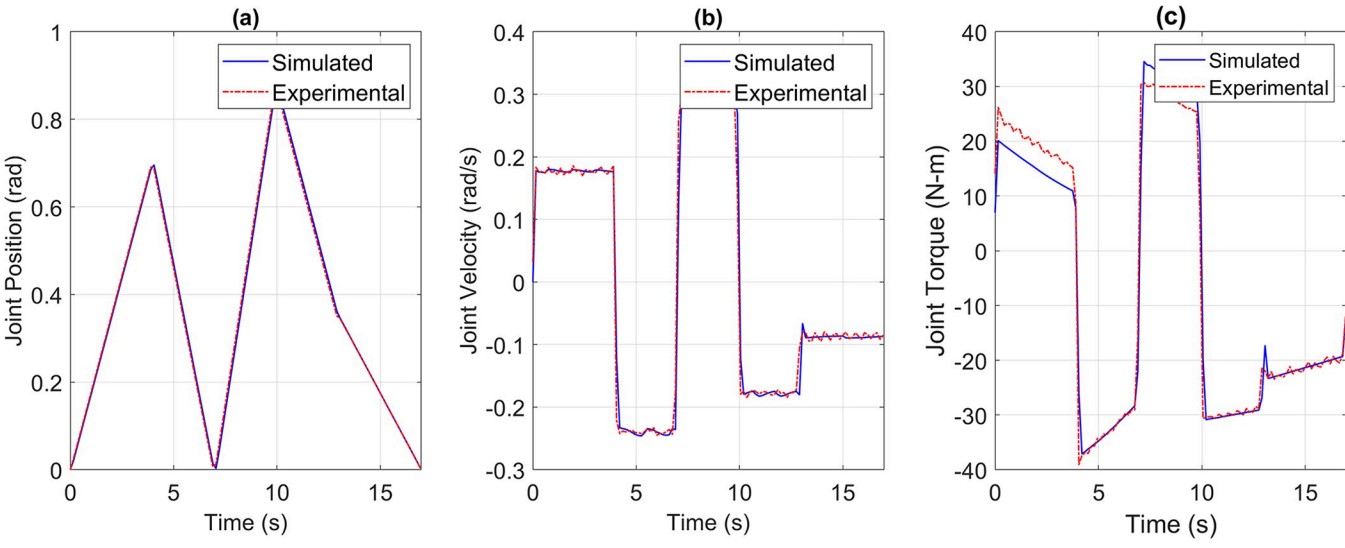

**Fig 25.** Simulated versus Experimental data of Joint 4's (a) Position, (b) Velocity, and (c) Torque.

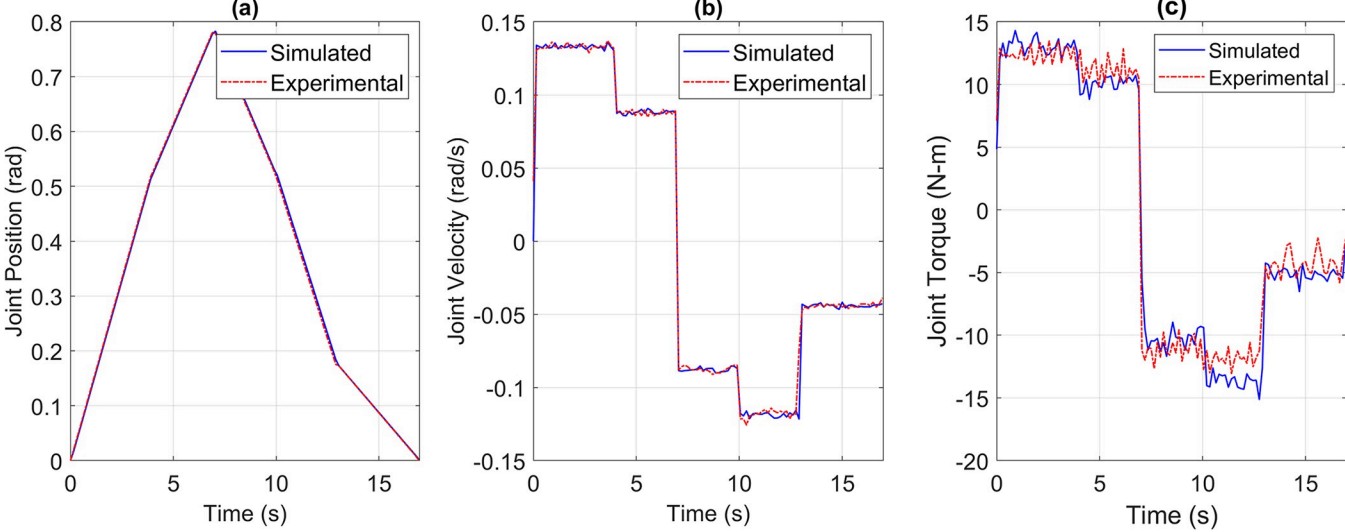

**Fig 26.** Simulated versus Experimental data of Joint 5's (a) Position, (b) Velocity, and (c) Torque.

**Table 6. Pearson's correlation coefficient between simulated and actual torque profiles of all joints for pick and place trajectory.**

| Parameter | Joint 1 | Joint 2 | Joint 3 | Joint 4 | Joint 5 |
|---|---|---|---|---|---|
| Correlation Coefficient | 0.9966 | 0.9872 | 0.9882 | 0.9888 | 0.9895 |

**Table 7. Root Mean Squared Error of joint space position (rad), velocity (rad/s), and torque (Nm) trajectories calculated using the simulated and measured trajectories of physically developed 5 DOF manipulator.**

| Trajectory | Joint 1 | Joint 2 | Joint 3 | Joint 4 | Joint 5 |
|---|---|---|---|---|---|
| Position (rad) | 0.0018 | 0.0086 | 0.0042 | 0.0091 | 0.0046 |
| Velocity (rad/s) | 0.0062 | 0.0223 | 0.0134 | 0.0238 | 0.0125 |
| Torque (Nm) | 1.8585 | 5.4244 | 2.3571 | 3.7471 | 1.5893 |

**Table 8. Standard deviation and repeatability of 5-DOF robot measured and calculated as suggested in ISO 9283:1998 for five different positions.** All values are in mm.

| Robot | Position 1 | | Position 2 | | Position 3 | | Position 4 | | Position 5 | |
|-------|------|------|------|------|------|------|------|------|------|------|
| | S | Rp | S | Rp | S | Rp | S | Rp | S | Rp |
| 5-DOF | 0.0018 | 0.0080 | 0.0039 | 0.0163 | 0.0062 | 0.0260 | 0.0107 | 0.0503 | 0.0030 | 0.0138 |

Where '$x_i$' is the measured value in i$^{th}$ iteration, '$\bar{x}$' is the mean position of all iterations, 'n' is the number of iterations, and '$\bar{l}$' is then the mean positional repeatability according to ANSI/RIA R15.05 standard [40]. Moreover, the standard deviation 'S' is then calculated as:

$$S = \sqrt{\sum_{i=1}^{n} \frac{(l_i - \bar{l})^2}{n-1}} \tag{10}$$

Lastly, the positional repeatability, 'Rp', is defined and calculated as

$$Rp = \bar{l} + 3S \tag{11}$$

Table 8 shows the Standard Deviation and the Positional Repeatability for each position. The worst value of the repeatability achieved in this experimentation is claimed as the repeatability of the developed robot prototype. Comparison with commercially available lightweight robots establishes that this value of repeatability is acceptable for robotic manipulators. The worst-case value of repeatability, 0.0503 mm, of our developed robotic manipulator is equivalent to commercially available lightweight robot Techman Robot TM5 [41].

Overall, the developed simulation tool demonstrates its potential for robot design and development, particularly in the selection of integrated drive joint subcomponents. Using the proposed scheme, the authors along with their research team have also developed a 6-DOF robot manipulator, with 1 m reach and 4 Kg payload capacity, shown in Fig 27, is in the testing phase. Fig 27 shows the configuration and hardware of the new manipulator.

## Section V: Conclusion and future work

This research underscores the benefits of a component-driven approach for designing and building lightweight robotic manipulators. By leveraging models of commercially available components within the design process, the gap between simulations and real-world systems narrows significantly. This can lead to eliminating or streamlining various steps in the development process, resulting in faster and more efficient creation of lightweight robotic manipulators.

The presented framework demonstrates the effectiveness of this approach through the modeling and analysis of a physical prototype of a 5-DOF lightweight robotic manipulator. Notably, the experimental evaluation of the developed robotic system yielded impressive results. The positional repeatability, assessed in accordance with ISO 9283:1998, demonstrated a repeatability of 0.05 mm, a performance level comparable to the Techman Robot TM5, a commercially available lightweight robot.

Following hardware experiments, where the same integrated drive joints and robot structure were manufactured and tested along predetermined trajectories, results showed close alignment between the simulated model and the actual position and velocity readings. However, a noticeable difference was observed in the required torque compared to the simulation. This discrepancy can be attributed to factors not included in the model, such as friction from the strain wave gears and potential misalignments introduced during assembly.

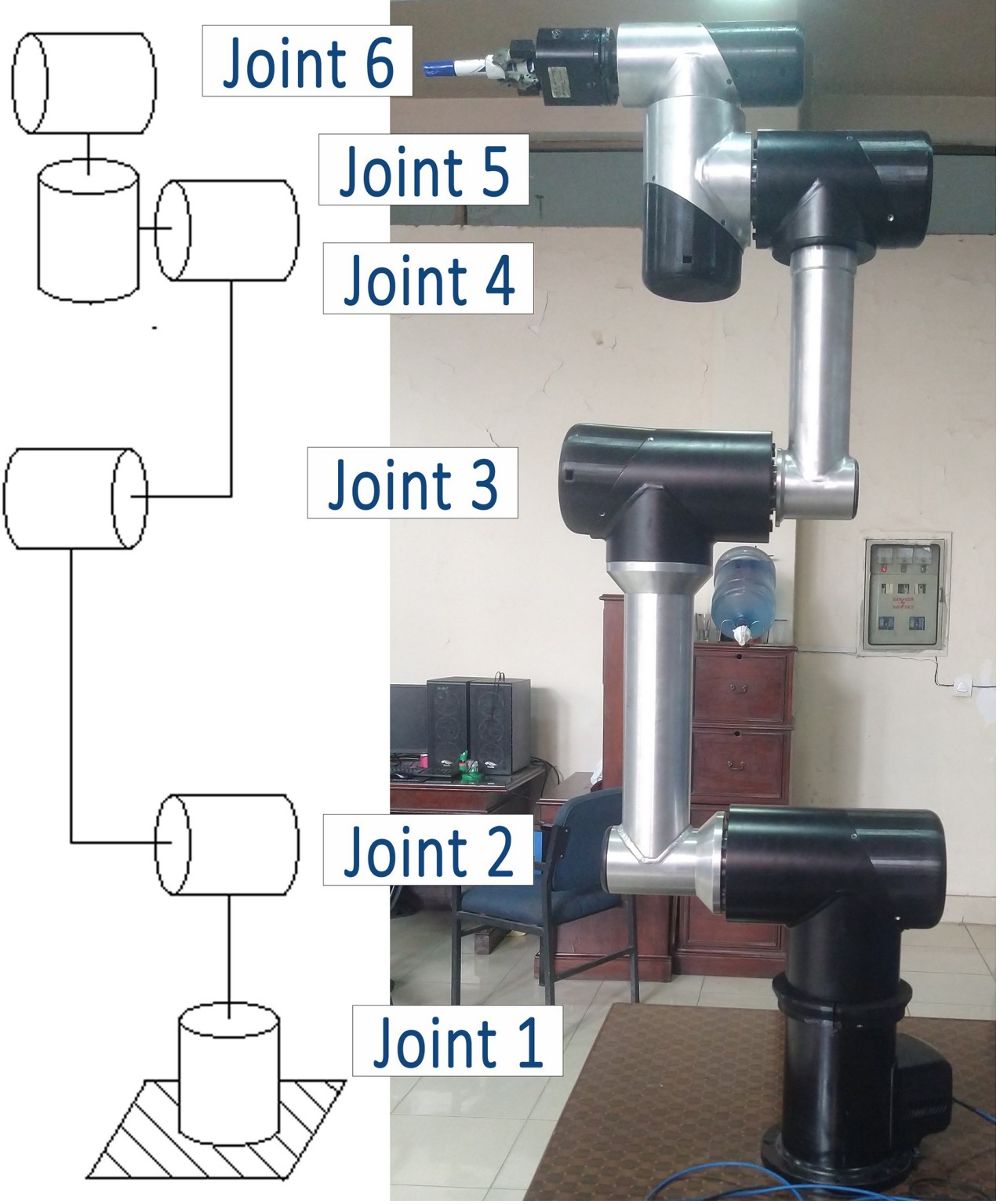

**Fig 27.** Newly developed 6 DOF Robot's FBD (Left) and its hardware (Right).

Encouragingly, the magnitude of this torque difference remained minor compared to the joints' nominal torque capacity.

Our study presents a simulation tool and framework with the potential to significantly transform the robot manufacturing industry. By allowing developers in educational research, robotics startups, and related fields to simulate manipulators using real-world component models, it offers substantial savings in time, effort, and investment. This is particularly valuable in today's fast-paced and resource-constrained environment.

Future work will focus on several key areas: refining the simulation tool for even greater accuracy, expanding the library of simulated models to encompass a wider range of components from diverse manufacturers, and tackling the complexities of unmodeled dynamics. By continuously enhancing the framework's accuracy, reliability, and user-friendliness, we aim to solidify its position as a transformative tool within the robot manufacturing industry.

## Acknowledgments

We would like to thank Dr. Ali Raza, Principal Investigator (PI) of Human Centered Robotics Lab UET Lahore for his unwavering support and guidance.

## Author Contributions

**Conceptualization:** Muhammad Ahsan.

**Data curation:** Muhammad Rzi Abbas.

**Formal analysis:** Muhammad Rzi Abbas.

**Investigation:** Muhammad Rzi Abbas, Muhammad Ahsan.

**Methodology:** Muhammad Rzi Abbas, Muhammad Ahsan.

**Project administration:** Jamshed Iqbal.

**Resources:** Jamshed Iqbal.

**Software:** Muhammad Ahsan.

**Supervision:** Muhammad Ahsan.

**Validation:** Jamshed Iqbal.

**Visualization:** Jamshed Iqbal.

**Writing – original draft:** Muhammad Rzi Abbas.

**Writing – review & editing:** Jamshed Iqbal.

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
