## [Decision Letter · Decision Letter 0]

20 Nov 2023

PONE-D-23-31418Experimental Development of Lightweight Manipulators with Improved Design Cycle Time that leverages off-the-shelf Robotic Arm ComponentsPLOS ONE

Dear Dr. Iqbal,

Thank you for submitting your manuscript to PLOS ONE. After careful consideration, we feel that it has merit but does not fully meet PLOS ONE’s publication criteria as it currently stands. Therefore, we invite you to submit a revised version of the manuscript that addresses the points raised during the review process.

We look forward to receiving your revised manuscript.

Kind regards,

Omer Saleem, Ph.D.

Academic Editor

PLOS ONE

3. We note that Figure(s) 2, 5, 6, 7, 8, 9, 10, 18 and 26 in your submission contain copyrighted images. All PLOS content is published under the Creative Commons Attribution License (CC BY 4.0), which means that the manuscript, images, and Supporting Information files will be freely available online, and any third party is permitted to access, download, copy, distribute, and use these materials in any way, even commercially, with proper attribution. For more information, see our copyright guidelines: http://journals.plos.org/plosone/s/licenses-and-copyright.

a. You may seek permission from the original copyright holder of Figure(s) 2, 5, 6, 7, 8, 9, 10, 18 and 26 to publish the content specifically under the CC BY 4.0 license. 

4. Please amend the manuscript submission data (via Edit Submission) to include author Muhammad Ahsan.

5. Please amend your authorship list in your manuscript file to include author Ahsan Ali.

Additional Editor Comments:

The authors have presented a lightweight robotics design framework in this paper. The study emphasizes the advantages of using a component-driven design methodology for creating lightweight robotic manipulators. It utilizes a 5 Degrees-of-Freedom (DOF) robotic manipulator (cobot prototype) for testing purposes, which has yielded interesting results. The paper presents an innovative idea that is supported by interesting results.

The paper was reviewed by two (02) reviewers. According to the reviewer's comments, the paper cannot be accepted in its present form. However, I am willing to consider a revised version of this manuscript. I suggest that the authors follow the manuscript guidelines available on the PLOS ONE website and re-edit the manuscript draft to comply with the official template of PLOS One. As per the reviewer's comments, the literature review should be extended, the efficacy of other model-free control schemes (PID, PD, etc) must also be rigorously analyzed for this application, and additional simulations can be added to examine the controller's robustness against step/impuslive disturbances. The detailed reviewer's comments are included.

Reviewers' comments:

Reviewer's Responses to Questions

**Comments to the Author**

1. Is the manuscript technically sound, and do the data support the conclusions?

Reviewer #1: Yes

Reviewer #2: Yes

2. Has the statistical analysis been performed appropriately and rigorously? 

Reviewer #1: Yes

Reviewer #2: Yes

3. Have the authors made all data underlying the findings in their manuscript fully available?

Reviewer #1: Yes

Reviewer #2: Yes

4. Is the manuscript presented in an intelligible fashion and written in standard English?

Reviewer #1: Yes

Reviewer #2: Yes

5. Review Comments to the Author

Reviewer #1: The paper proposed a design frame for lightweight robotics. The research highlights the benefits of a component-driven approach in designing lightweight robotic manipulators. The authors streamlined the development process by integrating commercially available components into the design process. The study focused on integrated drive joints in a 5-Degree of freedom (DOF) robotic manipulator, achieving impressive results. The proposed design strategy has the potential to transform the robotics industry, offering time and cost savings for developers in various sectors. The paper is well-written, however, there are some concerns that need to be addressed.

1. The authors should delve into existing literature and cite papers that address the reduction of the conventional design cycle in robotics.

2. Although a PI controller was utilized for the motor, the authors did not explore alternative controllers like PID or PD. It remains unclear why other controller types were not considered. Further investigation into the choice of the PI controller and the omission of PID or PD controllers could provide valuable insights into the motor control strategy adopted in this study. The authors can provide justification through simulation results as well.

3. The authors presented simulation and experimental results, yet they should further validate the design approach's feasibility by exploring additional scenarios, particularly under diverse disturbance conditions. Demonstrating the system's performance in the face of various disturbances would enhance the comprehensiveness and robustness of their findings.

4. The authors didn’t follow the formatting guidelines of the journal available on the Plos One website.

5. The references are not formatted according to the “International Committee of Medical Journal Editors (ICMJE)”. The reference should include all the information, such as the month of publication, and place in case of a conference.

6. Please report the values in SI units in Table 1.

Reviewer #2: Comments:

I strongly recommend the publication of this document due to its notable contributions to the field of lightweight robotics. The practical framework proposed by the authors offers a valuable solution to the longstanding challenges of high R&D costs and extended design cycles, providing a clear pathway for the development of lightweight industrial manipulators.

The case study involving a 5 Degrees-of-Freedom (DOF) cobot prototype serves as a compelling illustration of the framework's effectiveness. The achieved 40% reduction in design cycle time, coupled with a marginal 2.5% deviation in real-time performance, demonstrates the practicality and cost-effectiveness of the approach. This kind of tangible impact is crucial for the robotic industry, particularly for emerging startups and educational institutions with limited resources.

The document's thorough exploration of the experimental evaluation, including positional repeatability and addressing unmodeled dynamics, enhances its credibility. The authors showcase a deep understanding of real-world challenges, contributing valuable insights to the scientific community.

Furthermore, the foresight exhibited in envisioning the proposed simulation tool as a potential game-changer adds an innovative dimension to the document. The tool's potential to revolutionize the industry, coupled with its utility for developers in diverse settings, positions this work as not only a solution to existing problems but also a catalyst for future advancements.

In conclusion, the comprehensive nature of this document, from its well-executed case study to its forward-looking perspective, makes it a strong candidate for publication. Its potential to influence the field, particularly in making lightweight robotic development more accessible, aligns with the objectives of scientific dissemination and knowledge advancement.

Suggestions:

Authors might consider combining Sections II, III and IV as subsections under one section called Methodology?

6. PLOS authors have the option to publish the peer review history of their article (what does this mean?). If published, this will include your full peer review and any attached files.

Reviewer #1: No

Reviewer #2: No

---

## [Author Response · Author response to Decision Letter 0]

23 Jan 2024

Response to the reviewers file is attached for your kind review. Thanks a lot.

---

## [Decision Letter · Decision Letter 1]

3 Mar 2024

PONE-D-23-31418R1Experimental Development of Lightweight Manipulators with Improved Design Cycle Time that leverages off-the-shelf Robotic Arm ComponentsPLOS ONE

Dear Dr. Iqbal,

Thank you for submitting your manuscript to PLOS ONE. After careful consideration, we feel that it has merit but does not fully meet PLOS ONE’s publication criteria as it currently stands. Therefore, we invite you to submit a revised version of the manuscript that addresses ALL the points raised during the review process. Particularly, in your review, address issues related to the correctness of the equations and their units. As well as emphasizing the differences of this contribution with respect to the state of the art.

We look forward to receiving your revised manuscript.

Kind regards,

Carlos Alberto Cruz-Villar, Ph. D.

Academic Editor

PLOS ONE

Journal Requirements:

Reviewers' comments:

Reviewer's Responses to Questions

**Comments to the Author**

1. If the authors have adequately addressed your comments raised in a previous round of review and you feel that this manuscript is now acceptable for publication, you may indicate that here to bypass the “Comments to the Author” section, enter your conflict of interest statement in the “Confidential to Editor” section, and submit your "Accept" recommendation.

Reviewer #1: All comments have been addressed

Reviewer #2: All comments have been addressed

Reviewer #3: (No Response)

Reviewer #4: (No Response)

Reviewer #5: (No Response)

2. Is the manuscript technically sound, and do the data support the conclusions?

Reviewer #1: Yes

Reviewer #2: (No Response)

Reviewer #3: No

Reviewer #4: Partly

Reviewer #5: Yes

3. Has the statistical analysis been performed appropriately and rigorously? 

Reviewer #1: Yes

Reviewer #2: (No Response)

Reviewer #3: I Don't Know

Reviewer #4: N/A

Reviewer #5: Yes

4. Have the authors made all data underlying the findings in their manuscript fully available?

Reviewer #1: Yes

Reviewer #2: (No Response)

Reviewer #3: No

Reviewer #4: Yes

Reviewer #5: Yes

5. Is the manuscript presented in an intelligible fashion and written in standard English?

Reviewer #1: Yes

Reviewer #2: (No Response)

Reviewer #3: Yes

Reviewer #4: Yes

Reviewer #5: Yes

6. Review Comments to the Author

Reviewer #1: It is evident that each comment has been carefully considered, and appropriate modifications have been made to enhance the overall clarity and coherence of the paper.

Reviewer #2: (No Response)

Reviewer #3: 1)English should be revised in order to better understand the manuscript.

2) The state of the art on design methodologies must be reviewed and increased. Moreover, the manuscript's contribution needs to be clarified; there is no discussion of how this work differs from the state of the art. For example:

a) Padilla-García, E. A., Cervantes-Culebro, H., Rodriguez-Angeles, A., & Cruz-Villar, C. A. (2023). Selection/control concurrent optimization of BLDC motors for industrial robots. Plos one, 18(8), e0289717.

b) Cencen, A., Verlinden, J. C., & Geraedts, J. M. P. (2018). Design methodology to improve human-robot coproduction in small-and medium-sized enterprises. IEEE/ASME Transactions on Mechatronics, 23(3), 1092-1102.

c) Ronzoni, M., Accorsi, R., Botti, L., & Manzini, R. (2021). A support-design framework for Cooperative Robots systems in labor-intensive manufacturing processes. Journal of Manufacturing Systems, 61, 646-657.

d) Agarwal, P., Neptune, R. R., & Deshpande, A. D. (2016). A simulation framework for virtual prototyping of robotic exoskeletons. Journal of biomechanical engineering, 138(6), 061004.

e) Havard, V., Jeanne, B., Lacomblez, M., & Baudry, D. (2019). Digital twin and virtual reality: a co-simulation environment for design and assessment of industrial workstations. Production & Manufacturing Research, 7(1), 472-489.

f) E. A. Padilla-Garcia, A. Rodriguez-Angeles, J. R. ReséNdiz and C. A. Cruz-Villar, "Concurrent Optimization for Selection and Control of AC Servomotors on the Powertrain of Industrial Robots," in IEEE Access, vol. 6, pp. 27923-27938, 2018, doi: 10.1109/ACCESS.2018.2840537

g) Cusimano G, Casolo F. An almost comprehensive approach for the choice of motor and transmission in mechatronic applications: Torque peak of the motor. Machines. 2021;9(8):159.

3) More specifications must be provided including torque saturations, peak and steady-state current limits, and controller gains for stability.

4) In order to ensure clarity and accuracy, it would be greatly appreciated if a flowchart of the proposed methodology could be provided, highlighting the selected steps in yellow. This would aid in understanding the proposed methodology and facilitate the necessary review process.

5) At present, the stress analysis is carried out solely from a static point of view, with the maximum load applied to the final effector being the only consideration. However, the dynamic response of the system is influenced by the controller's gain tuning, and as such, the magnitude of values attained may surpass those observed in the static analysis.

6) Can you please clarify which method was used to find the optimal solution for the built robot? Also, could you explain why some of the methods available in the Matlab libraries were not utilized? It would be helpful if the wording could be more precise since the introduction states that the optimization stage was eliminated.

7) The proposed trajectories exhibit non-differentiability, thereby posing potential safety hazards for the cobot by introducing acceleration inconsistencies. This issue necessitates immediate attention and resolution to ensure the cobot's safe and optimal performance.

8) The specifics of the simulation should be included, e.g., the tuning of the gains of each degree of freedom, the sampling time, and the solver used to solve the differential equations.

9) A section should include all Solidworks parts and Matlab/Simscape files of the obtained design.

10) It is recommended to enhance the quality of all images. Additionally, capture images where only the robot is visible.

11) Each of the figures should be described and discussed in detail in the text.

12)Improve image quality. What is the maximum mass that the end effector can move? (Fig.27)

13) Could you kindly elaborate on what your goals are so that I can provide the most relevant insights? What are the criteria considered to design this robot?

14) Why use a correlation measure between data and not an error measure between the simulation and the physical prototype?

15) Clarify what they mean by satisfactory performance. It helps refer to equations or constraints of the system.

16) A section should include all Solidworks parts and Matlab/Simscape files of the obtained design.

17) Web pages should be omitted as references, instead sources from indexed journals should be sought.

18)t needs to be understood what initial design methodology needs to be improved. In addition, it is necessary to add where the changes are made concerning the proposed method.

19) Review the rest of the observations in the attached file.

Reviewer #4: The authors should add more quantitative data/analysis. The paper presents some quantitative results from simulations and experiments, but more data could strengthen the conclusions.

For example, providing specific statistics on the time/cost savings achieved using their method compared to traditional approaches, of course, based on explicit cites about the time/cost of traditional approaches.

The authors use simple PI control but, the justification for this implementation is missing. It is common to use this type of controller for BLDC motors, but it is imperative to justify, at least with standard or state-of-the-art information, about the advantages of PI over PD or PID controller for BLDC motors. On the other hand, the justification of using a PI controller can be done by results in the simulation of the system and get the justification that way.

It is imperative that the authors add data analysis between the simulation and experimental results regarding the trajectory tracking errors in terms of the error.

Additionally, add values of the tracking error in joint space between the reference and the measured values of the robot.

Reviewer #5: Comment #1: All equations should be numbered and referenced when needed.

Comment #2: The configured joints of the manipulator were configured with a damping constant of 2 Nm/deg/s, the usual SI damping units are Ns/m.

Comment #3: The iterated simulation using different integrated drive joints, does not yield an optimal joint as it’s mentioned.

Comment #4: Fig 8 does not show the Brushless DC Motor Drive (AC7) block and the connections used.

Comment #5: It is mentioned in the Abstract that the design cycle time is reduced by approximately 40% but no substantial evidence of this is explained during the results. Modifying the design approach does not inherently reduce the design cycle time since time saved in some areas may increase development time in other areas.

7. PLOS authors have the option to publish the peer review history of their article (what does this mean?). If published, this will include your full peer review and any attached files.

Reviewer #1: No

Reviewer #2: No

Reviewer #3: No

Reviewer #4: No

Reviewer #5: No

---

## [Author Response · Author response to Decision Letter 1]

27 Mar 2024

Response to reviewers file is attached for your kind review.

---

## [Decision Letter · Decision Letter 2]

22 Apr 2024

PONE-D-23-31418R2Experimental Development of Lightweight Manipulators with Improved Design Cycle Time that leverages off-the-shelf Robotic Arm ComponentsPLOS ONE

Dear Dr. Iqbal,

Thank you for submitting your manuscript to PLOS ONE. After careful consideration, we feel that it has merit but does not fully meet PLOS ONE’s publication criteria as it currently stands. Therefore, we invite you to submit a revised version of the manuscript that addresses the points raised during the review process.

Please be sure to fully address the comments of all reviewers, particularly those regarding the theoretical or methodological contribution, avoiding drawn on the experience of the designer. Please submit your revised manuscript by Jun 06 2024 11:59PM. If you will need more time than this to complete your revisions, please reply to this message or contact the journal office at plosone@plos.org. Please include the following items when submitting your revised manuscript:A rebuttal letter that responds to each point raised by the academic editor and reviewer(s). You should upload this letter as a separate file labeled 'Response to Reviewers'.A marked-up copy of your manuscript that highlights changes made to the original version. You should upload this as a separate file labeled 'Revised Manuscript with Track Changes'.An unmarked version of your revised paper without tracked changes. You should upload this as a separate file labeled 'Manuscript'.If applicable, we recommend that you deposit your laboratory protocols in protocols.io to enhance the reproducibility of your results. Protocols.io assigns your protocol its own identifier (DOI) so that it can be cited independently in the future. For instructions see: https://journals.plos.org/plosone/s/submission-guidelines#loc-laboratory-protocols. Additionally, PLOS ONE offers an option for publishing peer-reviewed Lab Protocol articles, which describe protocols hosted on protocols.io. Read more information on sharing protocols at https://plos.org/protocols?utm_medium=editorial-email&utm_source=authorletters&utm_campaign=protocols.

We look forward to receiving your revised manuscript.

Kind regards,

Carlos Alberto Cruz-Villar, Ph. D.

Academic Editor

PLOS ONE

Journal Requirements:

Reviewers' comments:

Reviewer's Responses to Questions

**Comments to the Author**

1. If the authors have adequately addressed your comments raised in a previous round of review and you feel that this manuscript is now acceptable for publication, you may indicate that here to bypass the “Comments to the Author” section, enter your conflict of interest statement in the “Confidential to Editor” section, and submit your "Accept" recommendation.

Reviewer #3: (No Response)

Reviewer #4: (No Response)

Reviewer #5: All comments have been addressed

2. Is the manuscript technically sound, and do the data support the conclusions?

Reviewer #3: No

Reviewer #4: Yes

Reviewer #5: Yes

3. Has the statistical analysis been performed appropriately and rigorously? 

Reviewer #3: Yes

Reviewer #4: I Don't Know

Reviewer #5: Yes

4. Have the authors made all data underlying the findings in their manuscript fully available?

Reviewer #3: Yes

Reviewer #4: Yes

Reviewer #5: No

5. Is the manuscript presented in an intelligible fashion and written in standard English?

Reviewer #3: Yes

Reviewer #4: Yes

Reviewer #5: Yes

6. Review Comments to the Author

Reviewer #3: Concerning comment 3, what is suggested to be validated is that the manufacturer's restrictions (torque saturations, peak and steady-state current limits,) are not violated in the simulation and the physical system.

Regarding comment 6, the author responds that they achieved an optimal solution through an iterative design refinement process. However, this process is subjective since there is no automatic process guiding the optimal solution; moreover, there is no guarantee of optimality. Furthermore, it would depend on the designer's expertise, so the validity of the time-saving would be questioned.

If the aim is to get a design, you may receive links larger than necessary and operate at relatively slow speeds, which may not reveal any issues. More reliable tests would involve displaying the design at ten times faster speeds, where dynamic effects can be seen and have a more significant impact. Moreover, you could suggest trajectories that work at resonance frequencies.

Until now, the only criterion to guide the validity of the design has been the robot's repeatability, which can be achieved with a well-tuned control and some possible feasible mechanical design solution.

With respect to comment 7, it is suggested that the signals obtained in the accelerations and jerk, both from the references and from the experimental signals, be reviewed.

Regarding comment 17, reference 12 does not mention anything about 10%, and reference 14 only discusses time ranges for three tasks. This bullet source from your reference 14 also needs to be clarified (for example, 2 to 10 weeks). It says design and coding. How much of that time is dedicated to design, and how much to coding? Likewise, the percentages reported in the stage designs in this article should be indicated in the references (11-14), but this is not the case.

It is still being determined where the percentages of the conventional design approach compared to the proposed design are obtained (Table 1). It also needs to be clarified how the authors guarantee time savings with the proposed methodology, given that no set of restrictions must be met to interrupt each of the design stages beyond having a correlation between the simulation and what is desired. For example, unlike what is proposed in reference 11, there is no talk of singularity analysis because there is no workspace analysis except doing a specific task.

Reviewer #4: Please double-check the manuscript for any grammatical errors, typos, or spelling mistakes, and ensure there are no extra spaces between words or sentences.

If the design and development stages are different from the general percentage values presented in Fig. 1, which ones are the particular percentage values for this implementation?

If Table 1 is one of the main results in this work, it should be addressed in a separate section of results and properly discussed, it should not be presented in the introduction because it demotivates the read of the article.

In Figs 13 to 17 are presented the references of each joint. I recommend adding the equations of the references proposed and also ad the acceleration profile of each joint.

The joint velocity profile in Figs 13 to 17 seems to be nondifferentiable. I recommend reading the book by Spong, M. W., Hutchinson, S., & Vidyasagar, M. (2020). Robot modeling and control. John Wiley & Sons. Chapter 7 is about path planning and explains why discontinuities in acceleration should be avoided.

Reviewer #5: (No Response)

7. PLOS authors have the option to publish the peer review history of their article (what does this mean?). If published, this will include your full peer review and any attached files.

Reviewer #3: No

Reviewer #4: No

Reviewer #5: No

---

## [Author Response · Author response to Decision Letter 2]

15 May 2024

Response to reviewers file is attached for your kind review. Thanks.

---

## [Editor Report · Decision Letter 3]

30 May 2024

Experimental Development of Lightweight Manipulators with Improved Design Cycle Time that leverages off-the-shelf Robotic Arm Components

PONE-D-23-31418R3

Dear Dr. Iqbal,

We’re pleased to inform you that your manuscript has been judged scientifically suitable for publication and will be formally accepted for publication once it meets all outstanding technical requirements.

Kind regards,

Carlos Alberto Cruz-Villar, Ph. D.

Academic Editor

PLOS ONE

Additional Editor Comments (optional):

I consider that the authors have revised their document in accordance with what was requested by the reviewers.
---

## [Editor Report · Acceptance letter]

13 Jun 2024

PONE-D-23-31418R3 

PLOS ONE

Dear Dr. Iqbal, 

I'm pleased to inform you that your manuscript has been deemed suitable for publication in PLOS ONE. Congratulations! Your manuscript is now being handed over to our production team.

Kind regards, 

on behalf of

Dr. Carlos Alberto Cruz-Villar 

Academic Editor

PLOS ONE